# Tax preference, financing constraints and enterprise investment efficiency—Experience, of China's enterprises investment

Liangliang Zhai[1], Yujing Feng[2]*, Fumin Li[3‡], Liping Zhai[4‡]

1 School of International Trade and Economics, Shandong University of Finance and Economics, Jinan, China, 2 Institute of Finance and Public Management, Anhui University of Finance and Economics, Bengbu, China, 3 Shandong Longquan Law Firm, Jinan, China, 4 Shandong Pingyin Rural Comercial Bank Co., Ltd, Jinan, China

☯ These authors contributed equally to this work.
‡ These authors also contributed equally to this work.
* fyj.ybyq@163.com

**Data Availability Statement:** All relevant data are within the article and its Supporting Information files.

## Abstract

This paper takes the 2014 pilot project of accelerated depreciation of fixed assets as a quasi-natural experiment, and builds a Propensity Score Matching–Difference in Differences (PSM-DID) model based on the data of Chinese listed companies from 2000 to 2019 to test the impact of tax preference on enterprise investment efficiency and its mechanism. The results show that the policy inhibits supported enterprises investment efficiency significantly. Heterogeneity analysis shows that the policy causes greater investment efficiency losses for small and medium-sized enterprises, non-state-owned enterprises and asset-heavy enterprises. The mechanism test found the reason why the policy eased financing constraints but inhibited investment efficiency in short-term. After a variety of robustness tests, the above basic conclusions are still valid. Although the accelerated depreciation policy of fixed assets is conducive to expanding the scale of investment, the incentive effect on investment efficiency is not obvious, and even shows a restraining effect. Given the existence of heterogeneity, the design of the policy should not only distinguish industries, but also pay attention to the differences between different enterprises in the same industry. Strengthening research and development (R&D) innovation and improving the matching mechanism between human capital and fixed investment will help give full play to the incentive effect of this policy. The research in this paper helps to deepen the understanding of the microeconomic effects of tax policy and identify the internal mechanism, which not only enriches the relevant literature, but also provides a reference for the government to better use tax policy to promote the high-quality development of enterprises.

## Introduction

At present, China's economic situation is complex, and uncertainty is rising year by year, and the improvement of investment efficiency has become the focus of scholars' research.

**Funding:** Anhui Province University Scientific Research Key Project (Humanities and Social Sciences), "Reform of Security Property Rights System, Financing Constraints and Domestic Value-Added Rate of Exports of Manufacturing Enterprises" (SK2021A0224). The key project of the National Social Science Foundation of China, "Analysis of the Political Economy of 'Reverse Globalization' and Research on China's Countermeasures" (18AGJ001). Youth Project of Natural Science Foundation of Anhui Province, China, "Economic Policy Uncertainty and Servicization of Manufacturing Enterprises: Theoretical Framework, Impact and Identification of Causal Effects" (2208085QG222). The funder played a role in the study design, analysis and decision to publish of the manuscript.

**Competing interests:** The authors have declared that no competing interests exist.

Investment has always been one of the "troikas" of China's economic growth, especially since the 2008 financial crisis, the Chinese economy has become more and more dependent on investment. Investment behavior is the process of converting assets with certain value such as monetary funds and manpower into capital, and the investment behavior involved in this thesis is the investment of enterprises in fixed assets. When an enterprise makes an investment, its purpose is theoretically to maximize the value of the enterprise, and such an investment is regarded as an efficient investment. However, in reality, the capital market has various problems such as information asymmetry, principal-agent, transaction costs, etc., so there will be inefficient investment behaviors, such as underinvestment and overinvestment.

Investment efficiency refers to the ratio between the effective results obtained by enterprise investment and the amount of input consumed or occupied, that is, the proportional relationship between the income and expenses, output and input of enterprise investment activities. Investment efficiency is a measure of how effectively an enterprise allocates scarce resources to investment projects and converts investment opportunities into actual investment [1]. In the case of low investment efficiency, it is difficult for enterprises to convert a large number of high-return investment opportunities into actual investment [2]. Improving the efficiency of corporate investment and avoiding invalid investment is an effective way for China to further deepen the supply-side structural reform.

Fazarri et al (1988) [3] put forward the financing constraint hypothesis, that is, in an imperfect capital market, information asymmetry, agency problems and related transaction costs make the internal and external financing costs of enterprises different, resulting in their external financing being constrained, which creates a situation that makes the business significantly dependent on internal funding. Under the circumstance of financing constraints, enterprises lack sensitivity to changes in capital costs, asset prices and investment opportunities, thus affecting the improvement of enterprise value. Due to the existence of transaction costs in the capital market, in a sense, all enterprises are faced with a certain degree of financing constraints, which is the resistance faced by enterprises in financing all their feasible investments. Tax preference is a common method used by the government to encourage enterprises to invest and upgrade technology by reducing costs and increasing cash flow, such as increasing total factor productivity [4] and cost-plus [5], and promoting research and development [6], Most scholars believe that tax preference can promote enterprise investment [7,8], where the core issue of this paper is how much of the increased enterprise investment is effective investment, that is, how the corporate investment efficiency changes under the incentives of policies.

The accelerated depreciation policy of fixed assets is a tax deferral, which is equivalent to providing an interest-free loan to enterprises to ease the financing constraints in the early stage of investment. Studies have shown that this policy can promote enterprise investment [8–10], and encourage R&D innovation [11–13], promote the upgrading of human capital [14] and so on. While this policy is expanding the scale of enterprise investment, it is worthwhile to further explore whether investment efficiency is also improving. Research on the factors affecting enterprise investment efficiency has been abundant, including financing constraints [15,16], industrial policy [17,18], monetary policy [19,20] and environmental uncertainty [21,22], etc., but there is little literature that focuses on the impact of this policy on enterprise investment efficiency. Based on this, this paper studies the impact of accelerated depreciation policies of fixed assets on corporate investment efficiency, which is a useful supplement to the existing literature.

Compared with the existing literature, the marginal contributions of this paper are mainly as follows: First of all, this paper studies the impact of preferential tax policies on enterprise investment from the perspective of investment efficiency, makes up for the relative lack of research in this field, and helps to understand the investment effects of preferential tax policies

more comprehensively. Secondly, the exogenous impact of the accelerated depreciation policy of fixed assets is used as a quasi-natural experiment, and a Difference in Differences model (DID model) is constructed for causal effect identification, which can effectively overcome the endogenous problems in the empirical test, thereby more accurately assessing the effect of tax preferential policies on enterprises investment efficiency. Thirdly, this paper identifies the mediating role of financing constraints and explores its possible mechanism in depth.

## Policy background and research hypothesis

In order to increase enterprises' enthusiasm to increase equipment investment, product upgrading and technological innovation, promote the transformation and upgrading of China's manufacturing industry, and improve the international competitiveness of the industry. In 2014, Chinese Ministry of Finance and the State Administration of Taxation jointly issued the "Notice on Improving the Corporate Income Tax Policy for Accelerated Depreciation of Fixed Assets" (Fiscal and Taxation [2014] No. 75), stipulating that the fixed assets newly purchased by enterprises in the six major industries after January 1, 2014 can shorten the depreciation period or adopt accelerated depreciation methods. The six major industries include: biopharmaceutical manufacturing, special equipment manufacturing, railway, shipbuilding, aerospace and other transportation equipment manufacturing, computer, communications and other electronic equipment manufacturing, instrumentation manufacturing, information transmission, software and information technology service industries. For enterprises, this policy can increase the deductible amount of enterprises in the initial stage of fixed asset investment and reduce the tax payable in the initial stage of investment. To a large extent, it will reduce the burden of taxes and fees, and then promote the high-quality development of enterprises.

The mechanism of the accelerated depreciation policy of fixed assets affecting corporate investment efficiency can be analyzed from the following two aspects (see Fig 1):

From the perspective of possible positive effects, this part of the tax saved by the accelerated depreciation policy of fixed assets is equivalent to providing an interest-free loan for them, which helps increase cash flow, reduce financing costs and ease financing constraints [23], especially for enterprises with greater financing constraints. At the same time, the implementation of the accelerated depreciation policy for fixed assets will send a benign signal to the market that the six major industries have good development prospects and greater investment potential, which is conducive to reducing the information asymmetry between credit banks and credit applicants; inspired by the motivation to chase high investment returns, private capital will flow to these industries to further ease the financing constraints faced by enterprises.

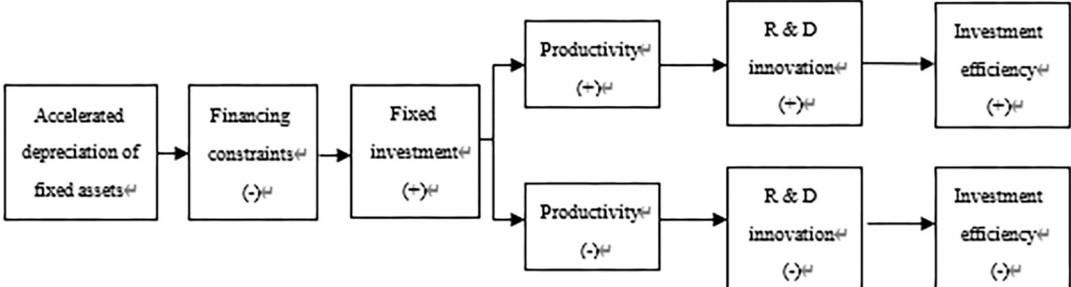

**Fig 1. The mechanism of the accelerated depreciation policy of fixed assets affecting the investment efficiency of enterprises.**

For enterprises with relatively scarce fixed assets, as the financing pressure eases, enterprises have enough capital to invest in fixed assets [10], which helps to alleviate the problem of insufficient investment. For enterprises that do not have underinvestment, there will be more funds to upgrade existing machinery and equipment and improve production technology. With the support of more advanced production technologies, enterprises' productivity will also increase; higher productivity can more effectively transform investment into output, and investment efficiency will increase accordingly. The accelerated depreciation of fixed assets has increased the sources of financing for enterprises, and coupled with the increase in productivity, it has provided financial guarantee and technical support for R&D and innovation. R&D investment is a special kind of investment, which has the characteristics of high risk and high return. The implementation of this policy has increased the enthusiasm of enterprises to carry out R&D and innovation. The income of R&D and innovation is much higher than that of ordinary investment, and it has the characteristics of increasing marginal income. The increase in the rate of return on investment will in turn help improve investment efficiency. The accelerated depreciation policy of fixed assets is a preferential tax policy for the six major industries, and its most direct effect is to reduce the tax burden and reduce the production cost of the key enterprises. Under the incentive of tax cuts, there will be positive changes in the financing constraints, productivity and R&D innovation of enterprises, which is conducive to the improvement of investment efficiency. Accordingly, the first research hypothesis is proposed:

Hypothesis 1: The accelerated depreciation policy of fixed assets will improve the investment efficiency of supported enterprises.

From the perspective of possible negative effects, for companies with less severe financing constraints or inadequate fixed assets, the tax preference that the accelerated depreciation policy of fixed assets brings to companies may not be fully and rationally utilized, and may even stimulate companies to make blind investments. However, the development of enterprise productivity lags behind the growth of investment scale, leading to the inability of investment to be fully transformed into output, resulting in the consequences of excessive investment. Under the guidance of market signals, resources flow to the six supported industries. The guiding role of the policy has replaced the guiding role of the market. Although companies in the six major industries are supported by policies, their productivity is not necessarily higher, and it is even possible that their productivity is lower than that of companies that are not supported by policies. In this way, resources will flow from industries with high productivity to industries with low productivity, distorted resource allocation and loss of efficiency, which will reduce the productivity of policy-supported enterprises [24,25]. When fixed investment increases and output is difficult to increase, the investment efficiency of enterprises will decrease. The accelerated depreciation policy of fixed assets is the state's external intervention in the economy, which will not only cause resource misallocation, but will also cause distortions in the capital-seeking-profit law, making enterprises less sensitive to investment opportunities and reducing investment efficiency. The accelerated depreciation policy of fixed assets will promote corporate investment and increase the scale of investment [10]. At the same time, the investment behavior of companies will also be affected by other companies in the same industry. Companies in the same industry make fixed investments under the incentives of policies, which will further stimulate the blind investment behavior of other companies, thereby forming a "herd effect", and the resource allocation efficiency and productivity of the entire industry will decline. Driven by the benefits of tax preference, companies have rent-seeking motives and may make additional fixed investments that are not actually needed, while reducing non-fixed investments. In this way, under external intervention, the investment structure of enterprises will change, and the alienation of investment behavior will also cause misallocation of

resources, which is not conducive to the improvement of productivity [26,27]. Companies that introduce a large number of machinery and equipment but cannot train highly skilled employees that match them in the same period will undoubtedly increase R&D costs, reduce R&D and innovation efficiency, and result in a loss of investment efficiency. Although the financing of accelerated fixed asset depreciation policy will promote enterprise investment, blind investment will produce large redundant costs, which will adversely affect productivity and R&D innovation, and the rate of return on investment will also decrease, which is not conducive to improving investment efficiency. If the company itself has sufficient fixed assets, considering the long period of fixed investment income, even if the tax saved by the policy is not overinvested, it will have the motivation to invest in financial assets with a short return period and quick results, which will promote financialization. But it is not conducive to the reproduction of enterprises and the improvement of productivity. Therefore, the second research hypothesis of this paper is proposed:

Hypothesis 2: The accelerated depreciation policy of fixed assets will reduce the investment efficiency of supported enterprises.

## Data and methodology

### Sample selection and data sources

This paper takes China's listed companies from 2010 to 2019 as a research sample. The data mainly comes from the China Stock Market Accounting Research Database (CSMAR) database, which is publicly available, has relatively complete corporate financial statements, and discloses corporate financial data every year, with good time continuity. In addition, the government also implemented preferential tax policies such as halving the income tax for small and micro enterprises to encourage the development of such enterprises. In order to prevent such policies from interfering with the conclusions, and to isolate the impact of accelerated depreciation of fixed assets on the efficiency of enterprise investment, this paper uses listed companies as a research sample. According to the actual situation and the research needs, the database has been processed as follows:

Excluding financial enterprises whose accounting standards are obviously different from those of ordinary enterprises and whose relevant indicators are not comparable. This paper focuses on the impact of the accelerated depreciation policy of fixed assets in 2014 on the investment efficiency of enterprises in the six major industries. In 2015, the policy was extended to four industries including light industry, textiles, machinery, and automobiles. In order to avoid policy interference, the four industries affected by the 2015 policy were excluded. In order to compare the changes in the investment efficiency of enterprises before and after the implementation of the policy, samples of enterprises established before 2014 were excluded. Exclude enterprises in ST (Special Treatment) or ST* status that are on the verge of delisting with poor reliability of relevant indicators. Finally, eliminate enterprises with missing values or obvious outliers.

### Model setting and variable selection

In order to test the impact of the accelerated depreciation policy of fixed assets on the investment efficiency of manufacturing enterprises, this paper uses the policy implemented in 2014 as a quasi-natural experiment to construct a Propensity Score Matching- Difference in Differences model (PSM-DID model).

Difference-in-difference (DID) has been a very popular method in recent years to assess the effects of regional policies. The basic idea is to take the regional policy as a quasi-natural experiment, differentiate the experimental group under the influence of the policy and the control

group not affected by the policy before and after the implementation of the regional policy, and then calculate the difference between the two groups of difference results, so as to obtain the net regional effect of the policy. However, a reasonable assessment of regional policy should first ensure that both the experimental and control groups are randomly selected, thereby avoiding the self-selection problem. In fact, the division of the experimental group and the control group is often not randomly selected, and there are different characteristics, which will cause the selectivity bias of the difference-in-difference method and further lead to endogeneity problems. Since the introduction of the accelerated depreciation policy for fixed assets has a great impact on heavy-asset enterprises, it is obviously not a random selection, which may lead to the problem of sample selection bias. Furthermore, to reasonably evaluate a regional policy, there needs to be a suitable control group, i.e., first, the experimental group and the control group are similar, and the experimental group is affected by the policy and the control group is not affected by the policy.

The propensity score matching method (PSM) is usually used to solve the problem of selection bias, and its basic idea is to form an approximate randomized experiment by constructing a counterfactual framework. The so-called counterfactual refers to observing the consequences of the experimental group without policy intervention through the control group, and then comparing the two results to eliminate the problem of selection bias, so as to obtain the true causal relationship. The PSM-DID method is to first use the PSM method to eliminate the selection bias in the sample, and then use the DID method to identify the causal effect.

The PSM-DID method has been widely recognized and applied. For example, Wang Zhiyong (2022) [28] used the PSM-DID method to evaluate the industrial efficiency of the revitalization policy of old industrial bases in Northeast China. Gong Maogang and Zhang Meijiao (2022) [29] used the PSM-DID method to study the positive impact of the "three rights separation" of contracted land and agricultural subsidies on agricultural mechanization. In addition, a large number of scholars such as Zhang Minglin and Li Huaxu (2021) [30] and Si Chunxiao (2021) [31] have applied the PSM-DID method to academic research, and obtained scientific and reasonable conclusions.

The settings of grouping dummy variables are as follows:

The grouping dummy variable is represented by "Treat". If the companies that belong to the six industries are classified as the experimental group, Treat = 1; if the companies that do not belong to the six industries are classified as the control group, Treat = 0. The setting of the staging dummy variable is as follows: the staging dummy variable is represented by "Post". If the statistical year is 2014 or later and the policy has been implemented, then Post = 1; if the statistical year is before 2014 and the policy has not been implemented, then Post = 0. The industries supported by the accelerated depreciation policy of fixed assets are clearly defined, the time interval from introduction to implementation is relatively short, and it is more difficult for companies to repurchase fixed assets in the short term. Therefore, from the perspective of individual enterprises, this policy can be regarded as an exogenous impact, which provides a good precondition for the application of the DID model. The basis of the Difference in Differences method for estimating policy effects is that the same policy only affects individuals in the experimental group at the same time, and does not affect individuals in the control group. The counterfactual thinking behind it is that the experimental group and the control group only differ in whether they are affected by the policy, and there is no systematic difference in other aspects. In order to eliminate the systematic differences between the experimental group and the control group, this paper first conducts propensity score matching before constructing the DID model, and re-searches the corresponding control group individuals for the individuals in the experimental group by means of neighbor matching. First, use the Logit model to estimate the propensity score [32]. The dependent variable of the model is whether the company

belongs to the six major industries supported by the accelerated depreciation policy of fixed assets (Treat). The independent variable is a set of control variables used for matching, and the control variables are introduced below. Then, the common support domain is divided according to the predicted propensity score value, and neighbor matching is performed on the common support domain to obtain a control group sample that is similar to the experimental group sample in other aspects. Finally, use the matched samples for DID analysis.

This paper constructs the following DID model:

$$Invt_{i,t} = \beta_0 + \beta_1 Treat_i \times Post_t + \alpha X + \vartheta_i + \mu_t + \varepsilon_{i,t} \tag{1}$$

In Formula (1), the subscripts i and t respectively represent the individual enterprise and the time year. The explained variable $Invt_{i,t}$ represents the degree of investment inefficiency of firm i in year t. The explanatory variable is the interaction term $Treat_i \times Post_t$. X is a series of control variables at the enterprise level. $\vartheta_i$ represents the firm's individual fixed effect. $\mu_t$ represents the year fixed effect. $\varepsilon_{it}$ represents the random disturbance term. The given error term ($\varepsilon$) is assumed to be normally distributed with zero mean value and constant variance [33,34]. $\beta_0$ is a constant term, and $\alpha$ is the coefficient set of the control variable $\beta_1$ is the parameter that this paper focuses on, and its symbol and value represent the direction and magnitude of the impact of the accelerated depreciation policy of fixed assets on the inefficiency of corporate investment. With reference to existing research, the control variables are selected as follows: Take the natural logarithm of the total assets at the end of year t to obtain the size of the enterprise $size_t$; company listing time $age_t$, that is, the number of years the company has been listed, which is equal to the statistical year minus the listing Year; corporate internal cash flow $cflow_t$, equal to the net cash flow from operating activities at the end of year t divided by total assets; corporate asset-liability ratio $lev_t$, equal to the total liabilities at the end of year t divided by total assets; the fixed assets ratio $tan_t$ is equal to the fixed assets at the end of year t divided by the total assets; the growth rate of the total assets of the enterprise $grow_t$ is equal to the total assets at the end of the year minus the total assets at the end of the previous year, and then divides the end of the previous year Total assets; enterprise asset return $roa_t$, equal to the net profit at the end of year t divided by total assets; enterprise book-to-market value ratio $mbra_t$, equal to the total assets at the end of year t divided by market value.

The calculation method of enterprise investment inefficiency $Invt_{i,t}$ refers to the existing literature [35,36], and based on the model of Richardson (2006), using ordinary least squares OLS method to estimate the investment scale of the enterprise, and then get the residual of the regression. The specific estimation model is:

$$\begin{aligned} Invest_{i,t} = \alpha_0 &+ \alpha_1 tobin_{i,t-1} + \alpha_2 lev_{i,t-1} + \alpha_2 cash_{i,t-1} + \alpha_3 age_{i,t-1} + \alpha_4 size_{i,t-1} + \alpha_5 return_{i,t-1} \\ &+ \alpha_6 invest_{i,t-1} + \sum industry + \sum year + \varepsilon_{i,t} \end{aligned} \tag{2}$$

In Formula (2), the dependent variable is the investment scale of the enterprise $Invest_{i,t}$, which is equal to the cash paid for the construction of fixed assets, intangible assets and other long-term assets at the end of t, plus the purchase of subsidiaries and other business units, minus the cash received from disposal of subsidiaries and other business units, and finally divide the above result by the total assets at the end of the period. All control variables in the model are one period lagging, including the following: Tobin's Q value $tobin_{i,t-1}$ is equal to the value of tradable shares at the end of t-1 plus the value of non-tradable shares, plus the book value of liabilities, and finally divide the above result by the total assets at the end of the period. The asset-liability ratio $lev_{i,t-1}$ is equal to the total liabilities at the end of t-1 divided by the total assets. Cash holdings $cash_{i,t-1}$, equal to t-1 monetary funds at the end of the year plus short-term investments, plus transactional financial assets, and finally divided by the total

assets at the end of the period. Time to market of an enterprise $age_{i,t-1}$, that is, the number of listing years in t-1, which is equal to the statistical year minus the listing year. Enterprise size $size_{i,t-1}$, which is equivalent to taking the natural logarithm of the total assets at the end of t-1. Stock return $return_{i,t-1}$, expressed by the return rate of individual stocks considering cash dividend reinvestment in t-1 year. the investment inefficiency of the enterprise at the end of year t, $Invt_{it}$, is expressed by the absolute value of the regression residual. Further, residuals greater than 0 indicate over-investment, recorded as $Over\_Invt_{it}$; residuals less than 0 indicate under-investment, recorded as $Under\_Invt_{it}$. The above-mentioned dual fixed-effects model of dummy variables of industry and year controls the industry effect and time effect, and effectively alleviates the endogenous problem of omitted variables.

## Descriptive statistics of main variables

The descriptive statistics of the main variables in this paper are shown in Table 1. There are a total of 18647 observations, of which 6709 are over-invested observations and 11,938 are

**Table 1. Descriptive statistics of main variables.**

| variable | Variables abbreviation | Variables definition | Observations | Average | Standard deviation | Minimum | Maximum |
|---|---|---|---|---|---|---|---|
| the degree of Enterprise investment inefficiency | Invt | The degree of inefficiency of business investment | 18647 | 0.0445 | 0.0524 | 0.00001 | 0.4717 |
| the degree of enterprise overinvestment | Over_Onvt | Excessive degree of corporate investment | 6709 | 0.0618 | 0.0765 | 0.000015 | 0.4717 |
| the degree of enterprise underinvestment | Under_Invt | Insufficient level of business investment | 11938 | 0.0348 | 0.0272 | 0.00001 | 0.2657 |
| Grouped dummy variable | Treat | Grouping dummy variable, which indicates whether it belongs to the industry supported by the accelerated depreciation policy of fixed assets, if it belongs to the industry, the value is 1, otherwise the value is 0 | 18647 | 0.4082 | 0.4915 | 0 | 1 |
| staged dummy variable | Post | Stage dummy variable, which indicates whether it is after the implementation year of the accelerated depreciation policy for fixed assets, if so, the value is 1, otherwise the value is 0 | 18647 | 0.5160 | 0.4998 | 0 | 1 |
| the size of the enterprise | Size | Enterprise size, which is the natural logarithm of total assets at the end of the year. | 18647 | 21.9157 | 1.1637 | 17.1219 | 27.4677 |
| company listing time | Age | The number of years the company has been listed, which is calculated by subtracting the listing year from the statistical year. | 18647 | 9.1892 | 5.9331 | 2 | 29 |
| corporate internal cash flow | Cflow | Internal business cash flow, which is calculated by dividing net cash flow from operating activities at the end of the year by total assets. | 18647 | 0.0486 | 0.0738 | -1.9377 | 0.4876 |
| corporate asset-liability ratio | Lev | The company's asset-liability ratio, which is calculated by dividing total liabilities by total assets at the end of the year. | 18647 | 0.3724 | 0.2268 | 0 | 1.1315 |
| the fixed assets ratio | Tan | The firm's fixed asset ratio, which is calculated as year-end fixed assets divided by total assets. | 18647 | 0.0436 | 0.0417 | 0 | 0.6773 |
| the growth rate of the total assets of the enterprise | Grow | The growth rate of the company's total assets. It is calculated by subtracting the total assets at the end of the previous year from the total assets at the end of the current year, and then dividing the total assets at the end of the previous year. | 18647 | 0.0106 | 1.0878 | -66.5353 | 0.9900 |
| enterprise asset return | Roa | Corporate return on assets, which is calculated by dividing year-end net profit by total assets. | 18647 | 0.0306 | 0.1142 | -8.7534 | 0.3999 |
| enterprise book-to-market value ratio | Mbra | The company's book-to-market ratio, which is calculated by dividing total assets by market value at the end of the year. | 18647 | 0.6149 | 0.2560 | 0 | 6.5459 |

under-invested observations. The degree of enterprise investment inefficiency (Invt) has an average value of 0.0445, a minimum value of 0.00001, and a maximum value of 0.4717, and there are large differences between different companies. The degree of enterprise overinvestment (Over_Invt) has an average value of 0.0618, a minimum value of 0.000015, and a maximum value of 0.4717. The degree of enterprise underinvestment (Under_Invt) has the average value of 0.0348, the minimum value of 0.00001, and the maximum value of 0.2657. The average value of the grouping dummy variable "Treat" is 0.4082, indicating that nearly 41% of the enterprises in the sample studied in this paper belong to the six major industries and will be affected by the policy, so the selection of the experimental group is relatively representative. The average value of the staging dummy variable "Post" is 0.5160, indicating that the period after the policy is implemented accounts for more than 51% of the entire sample period, and the selection of the sample period is relatively representative. The conditions of other variables in the model are also listed in Table 1.

The time trend of investment inefficiency is shown in Fig 2. The degree of investment inefficiency has shown a downward trend as a whole, that is, investment efficiency has continued to rise. During the period 2014–2015, the degree of inefficiency has increased significantly. The policy object studied in this paper is the accelerated depreciation policy of fixed assets implemented in 2014, which coincides with the second rising period of investment inefficiency. From the overall research sample, the overinvestment in the inefficient investment of enterprises is more serious than the underinvestment. During the period 2014–2015, the degree of over-investment rose sharply, while the degree of under-investment rose slowly.

## Empirical results

### Sample matching effect test

In this paper, the balance test before and after PSM matching is shown in Table 2. Before matching, there are systematic differences between some covariates of the control group and

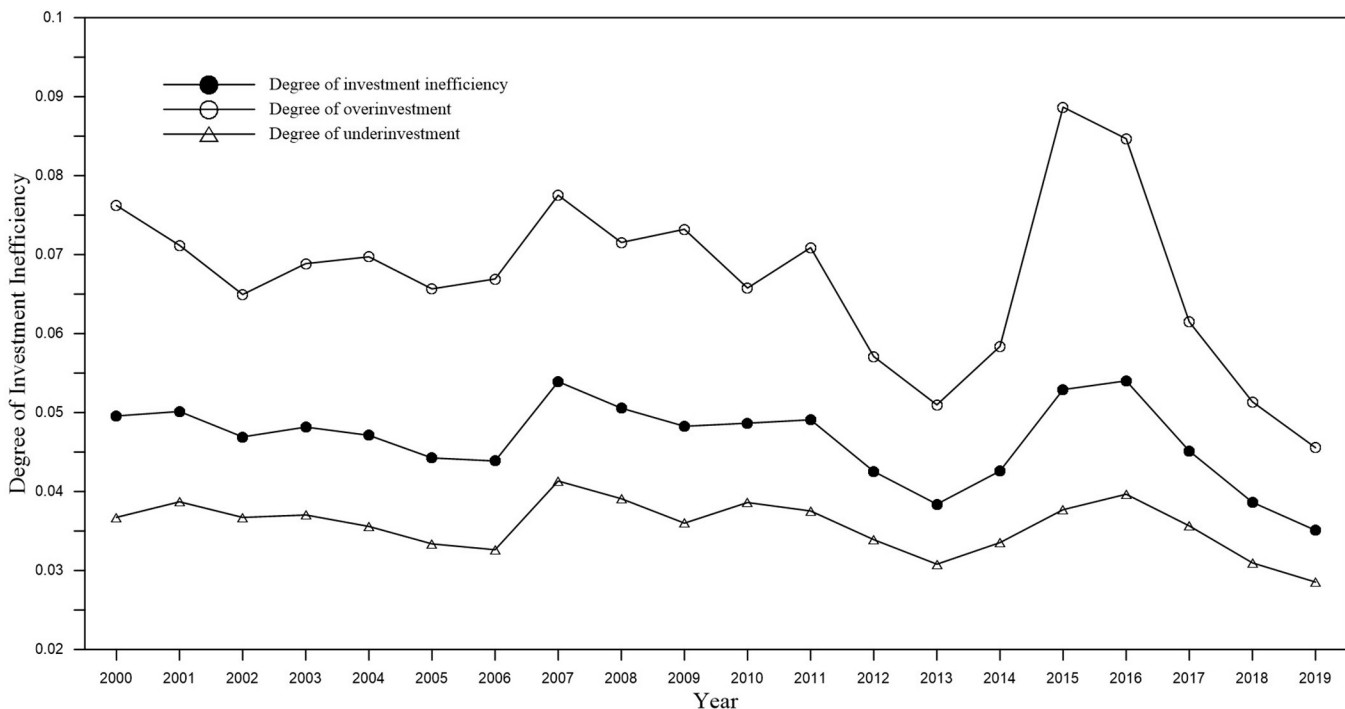

**Fig 2. Trend of investment inefficiency.**

**Table 2. Balance test before and after PSM matching.**

| Covariates | Covariates abbreviation | U/M | Mean | | % Deviation | t value | P value |
|---|---|---|---|---|---|---|---|
| | | | Experimental group | Control group | | | |
| the size of the enterprise | size | U | 21.891 | 22.008 | -9.8 | -5.95 | 0.000 |
| | | M | 21.891 | 21.873 | 1.5 | 0.87 | 0.386 |
| company listing time | age | U | 9.0615 | 9.1621 | -1.8 | -1.07 | 0.283 |
| | | M | 9.058 | 8.9573 | 1.8 | 0.98 | 0.328 |
| corporate internal cash flow | cflow | U | 0.0435 | 0.04939 | -8.1 | -4.92 | 0.000 |
| | | M | 0.04383 | 0.0425 | 1.8 | 1.03 | 0.303 |
| corporate asset-liability ratio | lev | U | 0.32375 | 0.40501 | -36.4 | -22.18 | 0.000 |
| | | M | 0.32377 | 0.32416 | -0.2 | -0.10 | 0.921 |
| the growth rate of the total assets of the enterprise | grow | U | 0.01525 | -0.00147 | 1.4 | 0.87 | 0.387 |
| | | M | 0.02273 | 0.03794 | -1.3 | -1.14 | 0.256 |
| enterprise book-to-market value ratio | mbra | U | 0.59183 | 0.65845 | -26.0 | -15.95 | 0.000 |
| | | M | 0.59096 | 0.59429 | -1.3 | -0.74 | 0.460 |
| enterprise asset return | roa | U | 0.03057 | 0.02805 | 2.1 | 1.37 | 0.170 |
| | | M | 0.03123 | 0.03215 | -0.8 | -0.46 | 0.647 |
| the fixed assets ratio | tan | U | 0.04354 | 0.04374 | -0.5 | -0.30 | 0.768 |
| | | M | 0.04352 | 0.04423 | -1.7 | -0.95 | 0.341 |

the experimental group, such as enterprise size "size", enterprise internal cash flow "cflow", enterprise asset-liability ratio "lev" and enterprise book-to-market value ratio "mbra". After the matching, there is no systematic difference between all the covariates, and the counterfactual idea of the DID model is satisfied. It can also be seen from the changes in the standard deviation of the covariates in Fig 3 that the covariates of the control group and the experimental group have achieved a better balance after matching.

## Analysis of DID results

**Benchmark inspection.** This paper uses the PSM method to re-match the control group and the experimental group, and then constructs a DID model for regression, and the results are shown in Table 3. Column (1) and column (2) both fix the individual enterprise effect and the time-year effect. Considering that companies in different industries may have different sensitivity to accelerated fixed asset depreciation policies, the model clusters at the industry level to eliminate differences between industries. There is only Treat×Post in column (1), and its coefficient is significantly positive at the 5% level. Column (2) adds a series of enterprise-level control variables on the basis of column (1). The coefficient of Treat×Post is still significantly positive at the 5% level, indicating that the accelerated depreciation policy of fixed assets can increase the investment inefficiency of supported companies and bring efficiency losses to corporate investment. Columns (3) and (4) have no fixed time-year effect, columns (5) and (6) have no fixed firm individual effects, and columns (7) and (8) are not clustered at the industry level. The coefficient of Treat×Post in column (3)-column (8) basically remains significantly positive, which also confirms the result of column (2) from the side. As a result, the accelerated depreciation policy of fixed assets will reduce the investment efficiency of supported companies, and Hypothesis 2 is confirmed. In consideration of avoiding and reducing endogenous problems as much as possible, factors that do not change with time and individual changes and differences in different industries should be controlled in the model, so follow-up studies are carried out on the basis of column (2).

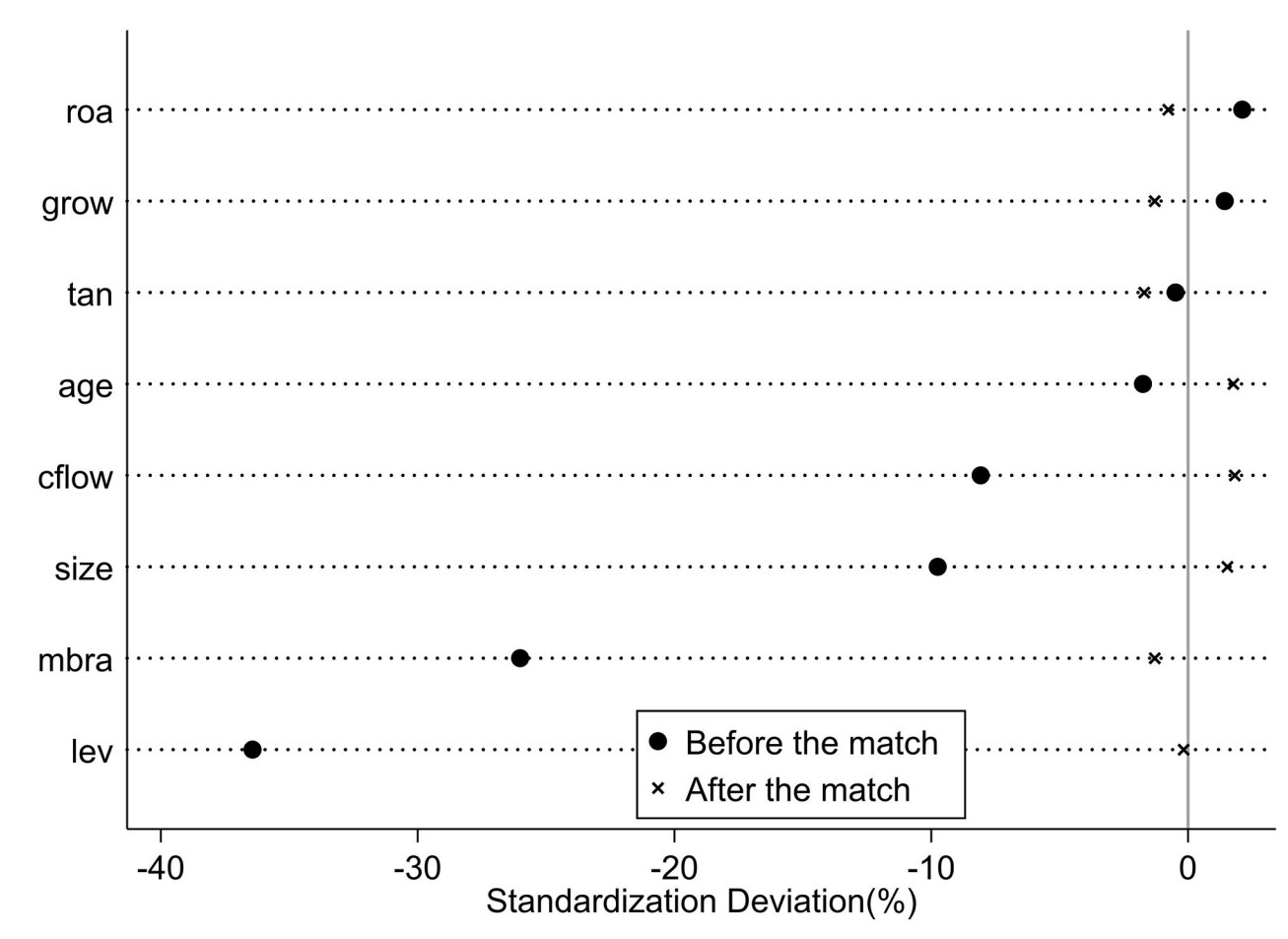

**Fig 3. Standardized deviation of covariates before and after matching.**

In terms of other control variables, the coefficient of enterprise size "size" is significantly positive at the 1% level. Large-scale enterprises generally have a large number of fixed assets, and the improvement of R&D and innovation capabilities often lags behind investment growth. As a result, fixed assets cannot be used fully and reasonably, and investment efficiency is reduced. The coefficient of a company's listing time "age" is significantly negative at the 1% level. With the extension of time for companies to go public, the matching of internal resources is more reasonable, the lag in productivity and R&D and innovation capabilities relative to investment growth has gradually disappeared, and the profitability of investment has been released, which will help reduce efficiency losses and improve investment efficiency. The coefficient of internal cash flow (cflow) is negative, which does not reach the level of significance. Investment is more sensitive to the company's internal cash flow. Abundant cash flow means that companies have sufficient internal financing. On the one hand, it can alleviate the financing constraints faced by enterprise investment, and the investment scale and efficiency will be improved. On the other hand, too much cash flow will limit the efficiency of capital use, but it is useless to improve investment efficiency. The coefficient of the corporate asset-liability ratio "lev" is significantly positive at the 1% level. The higher the debt-to-asset ratio, the greater the portion of the company's investment that comes from debt financing, so that

**Table 3. Benchmark regression results.**

| Variables | (1) | (2) | (3) | (4) | (5) | (6) | (7) | (8) |
|---|---|---|---|---|---|---|---|---|
| | the degree of Enterprise investment inefficiency (Invt) | the degree of Enterprise investment inefficiency (Invt) | the degree of Enterprise investment inefficiency (Invt) | the degree of Enterprise investment inefficiency (Invt) | the degree of Enterprise investment inefficiency (Invt) | the degree of Enterprise investment inefficiency (Invt) | the degree of Enterprise investment inefficiency (Invt) | the degree of Enterprise investment inefficiency (Invt) |
| Interaction item (Treat×Post) | 0.0053** | 0.0049** | -0.0051*** | 0.0040** | 0.0037* | 0.0050** | 0.0053*** | 0.0049** |
| | (0.0020) | (0.0023) | (0.0014) | (0.0016) | (0.0019) | (0.0022) | (0.0020) | (0.0024) |
| the size of the enterprise (size) | | 0.0122*** | | 0.0123*** | | 0.0061*** | | 0.0122*** |
| | | (0.0020) | | (0.0021) | | (0.0012) | | (0.0016) |
| company listing time (age) | | -0.0027*** | | -0.0028*** | | -0.0018*** | | -0.0027*** |
| | | (0.0003) | | (0.0003) | | (0.0001) | | (0.0003) |
| corporate internal cash flow (cflow) | | -0.0002 | | -0.0039 | | -0.0003 | | -0.0002 |
| | | (0.0084) | | (0.0102) | | (0.0077) | | (0.0078) |
| corporate asset-liability ratio (lev) | | 0.0153*** | | 0.0151*** | | 0.0228*** | | 0.0153*** |
| | | (0.0051) | | (0.0047) | | (0.0037) | | (0.0042) |
| the fixed assets ratio (tan) | | 0.0761** | | 0.0795*** | | 0.0825*** | | 0.0761*** |
| | | (0.0284) | | (0.0275) | | (0.0244) | | (0.0215) |
| enterprise book-to-market value ratio (mbra) | | -0.0180*** | | -0.0203*** | | -0.0160*** | | -0.0180*** |
| | | (0.0042) | | (0.0033) | | (0.0038) | | (0.0038) |
| enterprise asset return (roa) | | 0.0140** | | 0.0148** | | 0.0141** | | 0.0140* |
| | | (0.0066) | | (0.0064) | | (0.0059) | | (0.0075) |
| the growth rate of the total assets of the enterprise (grow) | | 0.0003 | | 0.0003 | | 0.0004** | | 0.0003 |
| | | (0.0002) | | (0.0002) | | (0.0002) | | (0.0004) |
| Individual fixed effect | YES | YES | YES | YES | NO | NO | YES | YES |
| Time fixed effect | YES | YES | NO | NO | YES | YES | YES | YES |
| Industry-level clustering | YES | YES | YES | YES | YES | YES | NO | NO |
| N | 18647 | 18647 | 18647 | 18647 | 18647 | 18647 | 18647 | 18647 |
| $R^2$ | 0.0262 | 0.0385 | 0.0009 | 0.0310 | 0.0256 | 0.0354 | 0.0262 | 0.0385 |
| F | 137.4 | 11391 | 13.12 | 27.09 | | | 20.98 | 15.33 |
| $Chi^2$ | | | | | 1898.58 | 182742.72 | | |

Note

*, **, and *** indicate significant at the 10%, 5%, and 1% levels, respectively, and the heteroscedasticity robust standard errors are in parentheses.

excessive debt leverage will increase the company's financing costs and operating risks. Even if the scale of investment increases, R&D and innovation capabilities are subject to excessive leverage, resulting in increased investment efficiency losses. The coefficient of the company's

fixed assets ratio "tan" is significantly positive at the 5% level. The higher the fixed asset ratio means the larger the scale of corporate investment. As the scale of investment increases, it is easy to cause excessive investment, which in turn reduces investment efficiency. The coefficient of corporate book-to-market value ratio (mbra) is significantly negative at the 1% level. The increase in the book-to-market value ratio means that compared with the current market value, the scale of the company's assets is too large, which can easily lead to excessive investment and reduce investment efficiency. The coefficient of corporate return on assets (roa) is significantly positive at the 5% level. The higher the rate of return on assets means the greater the benefits of investment, and companies are more motivated to continue investing. Investment efficiency depends not only on the return on investment, but also on the cost of investment. Driven by excessively high returns, companies are more likely to use debt leverage to make excessive investments, leading to increase in investment costs exceeding returns, which in turn leads to efficiency losses.

And the coefficient of the growth rate of total assets of an enterprise "grow" is positive but not significant. Under the circumstance of certain productivity and R&D and innovation capabilities, the excessive growth of corporate assets can easily lead to excessive investment and reduce investment efficiency; the scale of listed companies' assets is already large enough, and excessive investment costs will limit the growth rate of investment, thus the coefficient of "grow" did not reach the significance level.

The prosperity of China's economy is largely driven by investment, but the success of investment depends not only on size, but also on efficiency. Investment efficiency refers to the proportional relationship between the income obtained by an enterprise's investment and the resources it consumes or occupies. It is true that the implementation of the accelerated depreciation policy for fixed assets has achieved obvious effects, such as easing the financing constraints of enterprises, reducing leverage, and encouraging enterprise growth (Liu Xing et al., 2019 [10]; Tong Jinzhi et al., 2020 [37]; Zhong Guohui et al., 2021 [38]), and improving the cash flow of enterprises will promote the growth of investment scale (Xiong Bo and Du Jiaqi, 2020) [24]. This paper also finds that the accelerated depreciation policy of fixed assets will inhibit the improvement of investment efficiency of supported enterprises. After analyzing the positive and negative effects of the policy on the investment efficiency of enterprises, the empirical test shows that the negative effects will offset the positive effects, and the policy will have an adverse impact on the investment efficiency of supported enterprises as a whole. At present, China's economy is in a historical stage of further deepening supply-side structural reform and strengthening demand-side management. It is necessary to pay more attention to the improvement of investment efficiency while paying attention to the increase in the scale of enterprise investment. Looking at the world, the forces of anti-globalization are intensifying, and the new coronavirus epidemic is raging around the world, and the investment of enterprises is inevitably affected. For enterprises, although the accelerated depreciation policy for fixed assets does not change the total amount of depreciation and tax deduction in the useful life, it allows enterprises to accrue a large amount of depreciation in the current period when they purchase fixed assets, which can be regarded as a tax preference for deferred taxation. This policy releases the vitality of enterprises by reducing the current tax burden and easing financing constraints, such as promoting R&D investment (Li Haoyang et al., 2017) [13] and improving the structure of human capital (Liu Qiren and Zhao Can, 2020) [14]. It is undeniable that the policy will also have negative effects, such as inhibiting the improvement of the investment efficiency of supported enterprises, which is related to the effectiveness of enterprise investment, and will also affect other characteristics and behaviors of enterprises, and even offset the policy's effect to a certain extent. positive influence. Combined with the domestic and international situation, it is necessary to comprehensively analyze the advantages and

**Table 4. Test results of over-investment and under-investment.**

| | (1) | (2) | (3) | (4) |
|---|---|---|---|---|
| | Over_Invt | Over_Invt | Under_Invt | Under_Invt |
| Treat×Post | 0.0065 | 0.0080* | 0.0010 | 0.0008 |
| | (0.0040) | (0.0041) | (0.0010) | (0.0010) |
| Control variable | NO | YES | NO | YES |
| Individual fixed effect | YES | YES | YES | YES |
| Time fixed effect | YES | YES | YES | YES |
| Industry-level clustering | YES | YES | YES | YES |
| N | 6709 | 6709 | 11938 | 11938 |
| $R^2$ | 0.0353 | 0.0635 | 0.0543 | 0.0477 |
| F | 116.4 | 78.93 | 288.1 | 925.2 |

Note

*, **, and *** indicate significant at the 10%, 5%, and 1% levels, respectively, and the heteroscedasticity robust standard errors are in parentheses.

disadvantages of the accelerated depreciation policy of fixed assets for enterprises, so as to maximize the strengths and avoid weaknesses, and fully tap the incentive effect of the policy, which will help revitalize the market and promote the high-quality development of enterprises.

**Further expand the analysis.** This section further divides the inefficient investment of enterprises into over-investment and under-investment, in order to more specifically explore the impact of the accelerated depreciation policy of fixed assets on the investment efficiency of enterprises. The measurement indicators of over-investment and under-investment have been introduced in detail in the model setting and variable selection section above, and will not be repeated here. The explained variables in the model (1) are replaced with the degree of overinvestment of the enterprise (Over_Invt) and the degree of underinvestment (Under_Invt) respectively, and the DID analysis is performed again. The regression results are shown in Table 4. The explained variable in column (1) and column (2) is the degree of enterprise over-investment (Over_Invt). And the coefficient of Treat×Post is significantly positive at the level of 10%, indicating that the accelerated depreciation policy of fixed assets will increase the excessive investment of supported companies. The explained variable in column (3) and column (4) is Under_Invt of enterprise underinvestment. And the coefficient of Treat×Post is still positive but not significant, indicating that the accelerated depreciation policy of fixed assets will not aggravate the underinvestment of supported companies.

Therefore, the inhibitory effect of the accelerated depreciation policy of fixed assets on investment efficiency is mainly contributed by excessive investment. This policy did not aggravate the current underinvestment problem of enterprises, indicating that the enterprises did not use the tax savings saved in the current period for other purposes, but used the funds for investment, which was in line with the original intention of the implementation of the policy. On the whole, overinvestment in the Chinese economy is more serious than underinvestment, and the implementation of this policy has further aggravated the status quo to a certain extent. This policy aims to reduce the income tax burden in the early stage of investment, promote enterprises to speed up the renewal of machinery and equipment, and stimulate research and development innovation. The emergence of over-investment may be due to the low efficiency of resource allocation within the enterprise, or even the existence of resource misallocation, and the lag in the upgrading of human capital structure, which cannot match suitable human capital for newly increased fixed asset investment. Therefore, the investment made by the enterprise under the incentive of the policy can easily become the repeated investment, the

production potential cannot be fully tapped, the equipment upgrade cannot produce the technology spillover effect, and the productivity and R&D innovation of the enterprise cannot be improved accordingly.

**Robustness test.** *Parallel trend test.* The parallel trend hypothesis is the key hypothesis for using DID model to estimate the effect of a policy, which requires that the mean difference between experimental group and control group's explained variables remain consistent at different times before the policy has occurred, that is, the time trends of the two groups of explained variables remain consistent. The change trend of the investment inefficiency of the experimental group and the control group over time is shown in Fig 3. In the first five years of the accelerated depreciation policy for fixed assets implemented in 2014, the trends of the two groups were basically the same, while after the implementation of the policy in 2014, the trends of the two groups were significantly different. Which indicates that it was the impact of the policy led to changes in the explanatory variables of the experimental group.

Fig 4 is the judgment of the parallel trend hypothesis intuitively through graphics, and it's also necessary to use the regression coefficient method to conduct a more rigorous test of the hypothesis. On the basis of Eq (1) of the benchmark regression model, the interaction term between grouping dummy variable (Treat) and dummy variable of a certain or several years before the implementation of the policy is added, Formula (1) is extended to Formula (3):

$$Oserv_{it} = \beta_0 + \beta_1^1 Treat_i \times Before_1 + \beta_1^2 Treat_i \times Before_2 + \beta_1^3 Treat_i \times Before_3 + \beta_1^4 Treat_i \times Before_4 + \beta_1^5 Treat_i \times Before_5 + \beta_1 Treat_i \times Post_t + \alpha X + \vartheta_i + \mu_t + \varepsilon_{it} \quad (3)$$

The method of setting dummy variables for the years before the implementation of the policy in 2014 is as follows: Assuming that time t is the nth year before the implementation of the policy (n = 1,2,3,4,5), $Before_n = 1$, otherwise $Before_n = 0$. The significance of the coefficient of

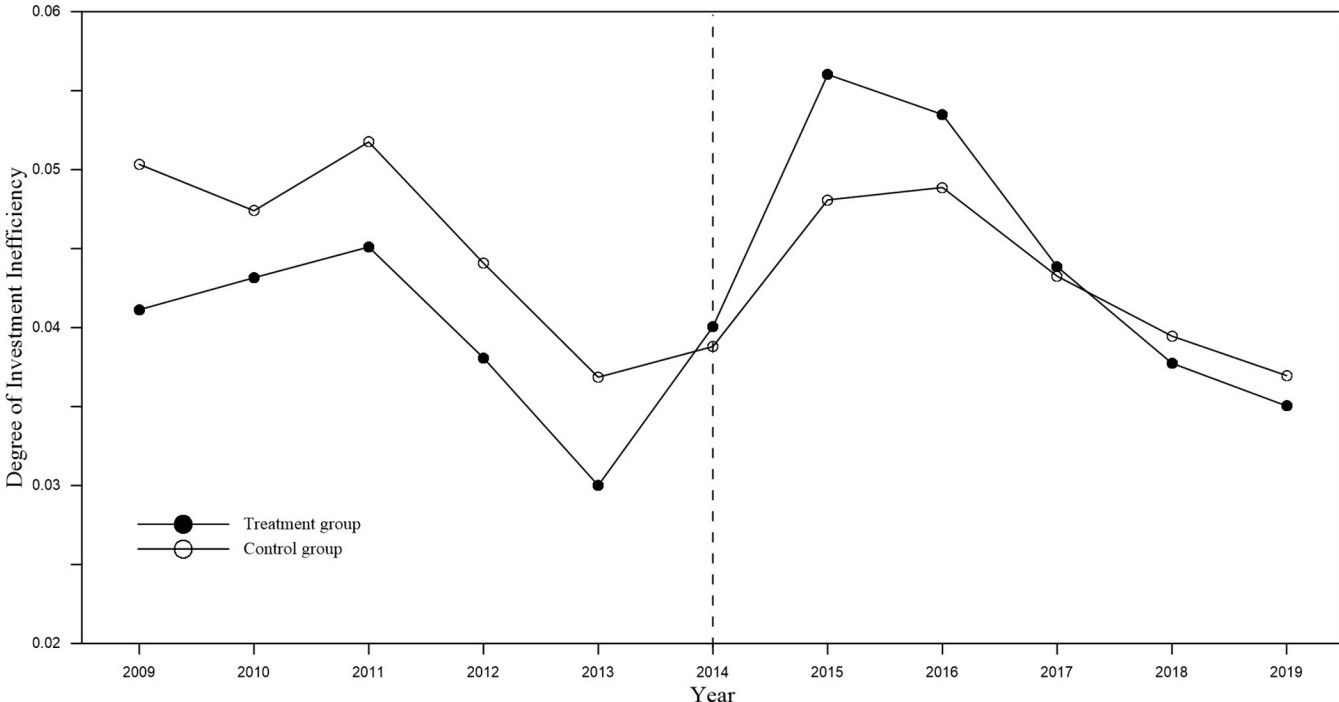

**Fig 4. The change trend of investment inefficiency of experimental and control group.**

the interaction term **Treat×Before_n** indicates the change trend between experimental group and control group before the policy is implemented. In addition, plot the regression coefficients and confidence intervals of **Treat×Post** and **Treat×Before_n** obtained by model (3), as shown by the solid line in Fig 5. Before the implementation of the policy, there is no systematic difference between the investment inefficiencies of the experimental group and the control group, that is, the DID model setting has passed the parallel trend hypothesis test.

*Placebo test.* One way to conduct a placebo test is to advance the year when the policy occurred, and re-do the Difference in Differences analysis with the data of the year before the policy occurred. Theoretically, any year before the policy occurs can be selected as the year of occurrence of the "virtual policy". If the "virtual policy" is found to have a significant effect, it means that even if the accelerated depreciation policy of fixed assets is not implemented, there will be differences between the experimental group and the control group.

Assuming that the policy was implemented in 2013, 2012, 2011, and 2010, set the staged dummy variables, and re-analyze the Difference in Differences model, the regression results are shown in Table 5. After adding the covariates, the estimated coefficients of Treat×Post2013, Treat×Post2012, Treat×Post2011, and Treat×Post2010 are not significant, indicating that the "virtual policy" does not exist. That is to say, the difference in the degree of investment inefficiency between the experimental group and the control group in this paper does come from the key support of the policy.

Another way to perform placebo testing is to repeat random sampling, by randomly assigning the experimental group and the control group to perform simulated regression. Although individual effects and year effects are fixed in the benchmark regression model and clustered at the industry level, the influence of unobservable factors cannot be completely ruled out. In order to confirm that the basic conclusions are not accidental, this part uses repeated random sampling methods for placebo testing. Randomly select a sample equal to the number of

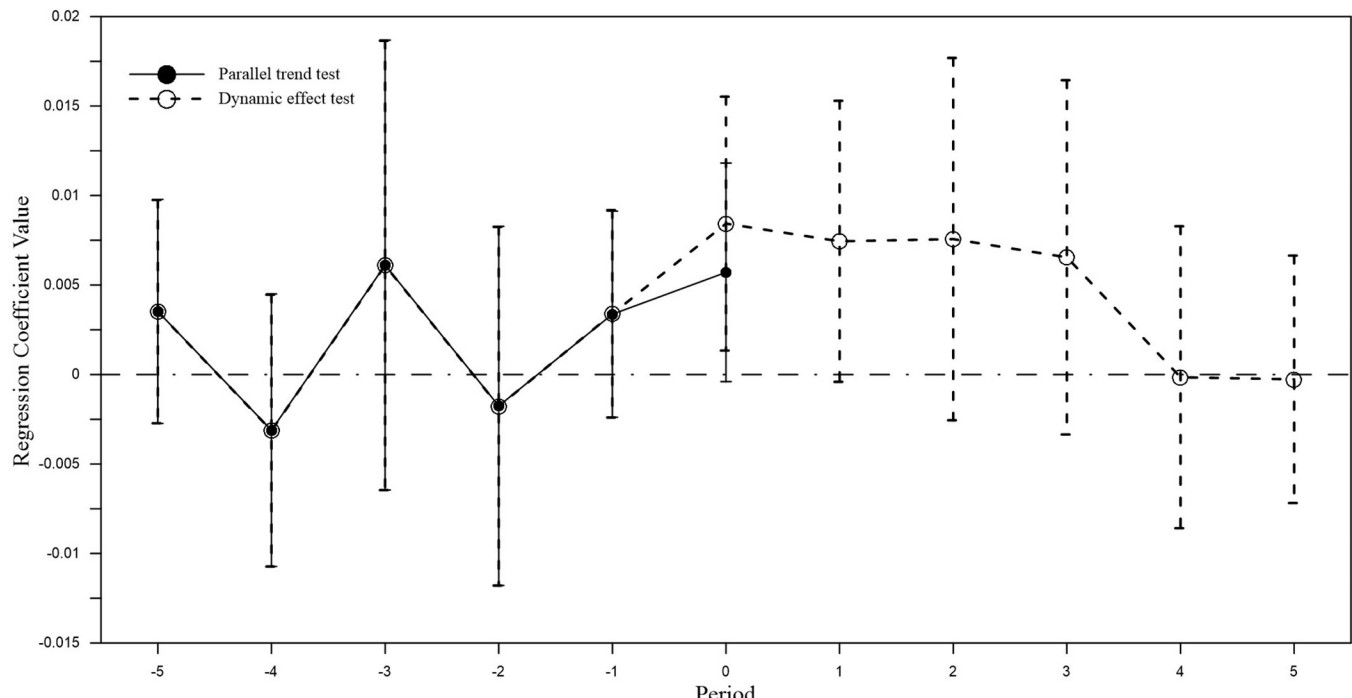

**Fig 5. Parallel trend test and dynamic effect test.**

**Table 5. Test results of early policy intervention years.**

| | (1) | (2) | (3) | (4) | (5) | (6) | (7) | (8) |
|---|---|---|---|---|---|---|---|---|
| | Invt | Invt | Invt | Invt | Invt | Invt | Invt | Invt |
| Treat×Post2013 | 0.0044** | 0.0031 | | | | | | |
| | (0.0021) | (0.0021) | | | | | | |
| Treat×Post2012 | | | -0.0004 | 0.0006 | | | | |
| | | | (0.0024) | (0.0022) | | | | |
| Treat×Post2011 | | | | | -0.0051 | -0.0045 | | |
| | | | | | (0.0037) | (0.0039) | | |
| Treat×Post2010 | | | | | | | 0.0031 | 0.0042 |
| | | | | | | | (0.0044) | (0.0050) |
| Other control variables | NO | YES | NO | YES | NO | YES | NO | YES |
| Individual fixed effect | YES | YES | YES | YES | YES | YES | YES | YES |
| Time fixed effect | YES | YES | YES | YES | YES | YES | YES | YES |
| Industry-level clustering | YES | YES | YES | YES | YES | YES | YES | YES |
| N | 18647 | 18647 | 18647 | 18647 | 18647 | 18647 | 18647 | 18647 |
| $R^2$ | 0.0257 | 0.0381 | 0.0256 | 0.0381 | 0.0257 | 0.0382 | 0.0256 | 0.0381 |
| F | 114.5 | 5235 | 86.59 | 3204 | 86.17 | 2249 | 84.65 | 1719 |

Note

*, **, and *** indicate significant at the 10%, 5%, and 1% levels, respectively, and the heteroscedasticity robust standard errors are in parentheses.

individuals in the actual experimental group as the virtual experimental group (that is, assuming Treat×Post = 1), and the rest as the virtual control group (that is, assuming Treat×Post = 0). DID analysis was performed according to the same model setting as the baseline regression, and random sampling was repeated 1000 times. The test result of this non-parametric random simulation is shown in Fig 5. The 1000 virtual regression coefficients obtained are mainly concentrated near the zero point, presenting a normal distribution approximately centered on 0, and most of the virtual coefficients do not reach the 10% significance level. The true estimated coefficient value of the policy effect is 0.0049, which is obviously an outlier in the distribution in Fig 6, indicating that there is a significant difference between the true estimated coefficient and the virtual regression coefficient.

The placebo test results based on the above two methods show that the basic conclusions of this article are not based on chance, and further verify the correctness of the DID model.

*Expected effect test.* Taking into account that in the year before the implementation of the policy studied in this paper, individual enterprises may form expectations related to the policy, so as to take countermeasures in advance. This policy is applicable to the depreciation of newly purchased fixed assets after January 1, 2014. If individual enterprises can expect it, they will tend to focus on investment in fixed assets after January 1, 2014, instead of purchasing new machinery and equipment in the year before the implementation of the policy. In this section, the sample whose statistical year is 2013 is removed, and the DID analysis is performed again. The regression results are shown in column (1) and column (2) of Table 6. The coefficient of Treat×Post in column (2) is still significantly positive, and the level of significance has increased, indicating that there is no expected effect of individual enterprises on policies. The basic conclusions of this paper are robust.

*Excluding the impact of expanding the scope of the policy pilot in 2015.* After the accelerated depreciation policy for fixed assets was implemented in the six major industries in 2014, four areas including light industry, textiles, machinery, and automobiles were further included in the pilot scope in 2015. In order to avoid confusion with the 2014 policy studied in this paper,

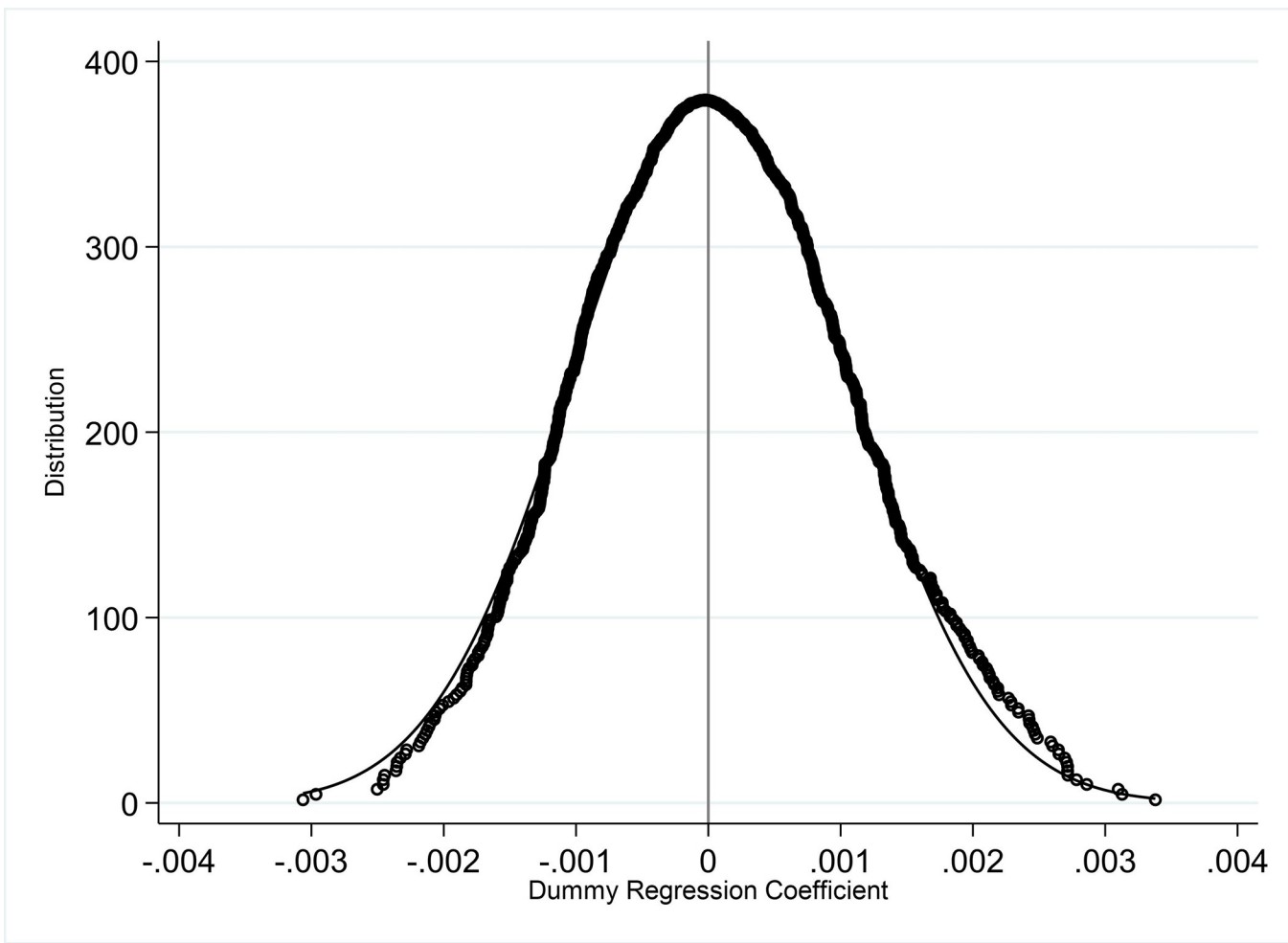

**Fig 6. Non-parametric random simulation results.**

**Table 6. Tests on expected effects, expanding the scope of the pilot program, and other policy effects.**

| | Exclude the 2013 sample | | Exclude new pilot samples in 2015 | | Exclude samples from the three northeastern provinces | |
|---|---|---|---|---|---|---|
| | **(1)** | **(2)** | **(3)** | **(4)** | **(5)** | **(6)** |
| | **Invt** | **Invt** | **Invt** | **Invt** | **Invt** | **Invt** |
| Treat×Post | 0.0069*** | 0.0064*** | 0.00616*** | 0.00408 | 0.0053** | 0.0051** |
| | (0.0020) | (0.0022) | (0.0022) | (0.0027) | (0.0021) | (0.0025) |
| Control variables | NO | YES | NO | YES | NO | YES |
| Individual fixed effect | YES | YES | YES | YES | YES | YES |
| Time fixed effect | YES | YES | YES | YES | YES | YES |
| Industry-level clustering | YES | YES | YES | YES | YES | YES |
| N | 17381 | 17381 | 14644 | 12122 | 17632 | 17632 |
| $R^2$ | 0.0266 | 0.0374 | 0.0285 | 0.0455 | 0.0263 | 0.0370 |
| F | 131.4 | 8089 | 220.5 | 117167 | 222.26 | 2450.23 |

Note

*, **, and *** indicate significant at the 10%, 5%, and 1% levels, respectively, and the heteroscedasticity robust standard errors are in parentheses.

this section removes the above-mentioned industries that were newly included in the pilot scope in 2015 and re-does the DID analysis. The regression results are shown in column (3) and column (4) of Table 6. The coefficient of Treat×Post in column (2) is still significantly positive, but the significance level has been reduced, indicating that the expansion of the policy pilot scope in 2015 did not have a substantial impact on the basic conclusions of this paper.

*Eliminate the confounding effects of other policies.* There are some other policies that may be mixed with the accelerated depreciation policy of fixed assets in 2014, causing confusion in the effect of the policy. An early policy closely related to the accelerated depreciation policy for fixed assets in 2014 that this paper focuses on is the accelerated depreciation policy of corporate income tax fixed assets and the value-added tax reform implemented in the three northeastern provinces (Heilongjiang, Liaoning, and Jilin) in 2004. In order to avoid the confusion of policy effects, this section removes the sample of enterprises from the three northeastern provinces and re-analyses the DID. The regression results are shown in column (5) and column (6) of Table 6. The coefficient of Treat×Post in column (2) is still significantly positive at the 5% level, indicating that the above-mentioned early policies did not have a substantial impact on the basic conclusions of this paper.

**Dynamic effect test.**   In order to further examine the time changes of the impact of accelerated depreciation policies on corporate investment efficiency, this section refers to the analysis method of Beck et al. (2010) to test the dynamic effects, and expands Eq (1) to Eq (3):

$$
\begin{aligned}
Oserv_{it} = {} & \beta_0 + \beta_1^1 Treat_i \times Before_1 + \beta_1^2 Treat_i \times Before_2 + \beta_1^3 Treat_i \times Before_3 + \beta_1^4 Treat_i \\
& \times Before_4 + \beta_1^5 Treat_i \times Before_5 + \beta_1 Treat_i \times Current_t + \beta_1^{1*} Treat_i \times After_1 \\
& + \beta_1^{2*} Treat_i \times After_2 + \beta_1^{3*} Treat_i \times After_3 + \beta_1^{4*} Treat_i \times After_4 + \beta_1^{5*} Treat_i \times After_5 \\
& + \alpha X + \vartheta_i + \mu_t + \varepsilon_{it}
\end{aligned} \tag{4}
$$

Where, the setting of the dummy variable $Before_n$ is the same as Formula (3). The dummy variable $Current_t$ represents the current year when the policy is implemented. Here, if the statistical year is 2014, the value is 1; otherwise, the value is 0. $After_n$ is the dummy variable of the year after the policy is implemented, and the setting method is similar to $Before_n$. The sign and significance of the coefficient of the interaction term $Treat \times After_n$ indicate the dynamic effect after the policy is implemented. The regression coefficient and its confidence interval are shown by the dotted line in Fig 5. The policy will still increase the investment inefficiency of supported companies in the first year after its implementation, but the effect will be significantly weakened. From the second year onwards, the effect of policies on the investment efficiency of enterprises has almost disappeared, and this phenomenon has also appeared in the trend of change in Fig 3. It may be because with the disappearance of the lag in enterprise R&D innovation, resource allocation, etc., the productivity effects and technology spillover effects of fixed investment gradually appear, and the negative effects of investment efficiency are also weakening. It can be seen that the accelerated depreciation policy of fixed assets does not have a strong continuity in suppressing the investment efficiency of supported enterprises, and the policy can only maintain a dynamic effect for one year.

**Mechanism test: The perspective of financing constraints.**   Combining theoretical analysis and benchmark regression results, the hypothesis that accelerated fixed asset depreciation policies will reduce the investment efficiency of supported companies is confirmed. As mentioned above, the most direct effect of the accelerated depreciation policy of fixed assets is to reduce financing costs and ease financing constraints. Therefore, this paper attempts to do further mechanism testing from the perspective of financing constraints. Construct the following

intermediary effect model:

$$Oserv_{it} = \beta_0 + \beta_1 Treat_i \times Post_t + \alpha X + \vartheta_i + \mu_t + \varepsilon_{it} \tag{5}$$

$$M_{it} = \beta_0 + \beta_1 Treat_i \times Post_t + \alpha X + \vartheta_i + \mu_t + \varepsilon_{it} \tag{6}$$

$$Oserv_{it} = \beta_0 + \beta_1 Treat_i \times Post_t + M_{it} + \alpha X + \vartheta_i + \mu_t + \varepsilon_{it} \tag{7}$$

For the selection of intermediary variables, referring to the practice of Hadlock and Pierce (2010), the SA index is used as a measure of financing constraints and the calculation formula is: $-0.737 \times Size + 0.043 \times Size^2 - 0.04 \times Age$.

The mechanism test results of the intermediary effect model are shown in Table 7. Where, columns (1)-column (3) show that the accelerated depreciation policy of fixed assets increases the investment inefficiency of enterprises by reducing financing constraints. Columns (4)-columns (6) show that this policy will increase the degree of overinvestment by reducing the financing constraints of enterprises. Columns (7)-column (9) show that the policy will not increase the degree of underinvestment by reducing the financing constraints of enterprises. The policy is mainly to increase the degree of overinvestment of enterprises, but has no significant impact on the degree of underinvestment; under the action of the policy, the financing constraints of supported enterprises are effectively alleviated, and they have more funds and motivation to make fixed investment. However, the production potential of new machinery and equipment cannot be fully explored under the existing resource allocation within the enterprise, resulting in excessive investment and loss of efficiency.

**Heterogeneity analysis.** Considering that the effect of accelerated depreciation of fixed assets on the investment efficiency of enterprises may differ in performance between different types of enterprises, this section examines the heterogeneity of scale, ownership, and asset structure. The regression results are shown in Table 8. Policies will significantly inhibit the investment efficiency of small and medium-sized enterprises, while the impact on large-scale enterprises will be insignificant. The possible reason is that large-scale enterprises generally have stronger operating capabilities, higher profits, better brand effects, higher accumulated

**Table 7. Mechanism test results.**

| | Degree of investment inefficiency Invt | | | Degree of overinvestment Over_Invt | | | Degree of underinvestment Under_Invt | | |
|---|---|---|---|---|---|---|---|---|---|
| | (1) | (2) | (3) | (4) | (5) | (6) | (7) | (8) | (9) |
| | Invt | SA | Invt | Over_Invt | SA | Over_Invt | Under_Invt | SA | Under_Invt |
| Treat×Post | 0.0049** | -0.1270** | 0.0026 | 0.0080* | -0.1270** | 0.0047 | 0.0008 | -0.1270** | 0.0006 |
| | (0.0023) | (0.0589) | (0.0022) | (0.0041) | (0.0589) | (0.0045) | (0.0010) | (0.0589) | (0.0010) |
| SA | | | -0.0183*** | | | -0.0309*** | | | -0.0017 |
| | | | (0.0022) | | | (0.0048) | | | (0.0010) |
| Control variables | YES | YES | YES | YES | YES | YES | YES | YES | YES |
| Individual fixed effect | YES | YES | YES | YES | YES | YES | YES | YES | YES |
| Time fixed effect | YES | YES | YES | YES | YES | YES | YES | YES | YES |
| Industry-level clustering | YES | YES | YES | YES | YES | YES | YES | YES | YES |
| N | 18647 | 18647 | 18647 | 6709 | 18647 | 6709 | 11938 | 18647 | 11938 |
| R² | 0.0385 | 0.2030 | 0.0615 | 0.0635 | 0.2030 | 0.0981 | 0.0477 | 0.2030 | 0.0484 |
| F | 11391 | 1185 | 3488 | 78.93 | 1185 | 663.1 | 925.2 | 1185 | 3165 |

Note

*, **, and *** indicate significant at the 10%, 5%, and 1% levels, respectively, and the heteroscedasticity robust standard errors are in parentheses.

**Table 8. Results of heterogeneity analysis.**

| | Large-scale enterprise | Small and medium-sized enterprises | State-owned enterprise | Non-state-owned enterprise | Asset-heavy enterprise | Asset-light enterprise |
|---|---|---|---|---|---|---|
| | (1) Invt | (2) Invt | (3) Invt | (4) Invt | (5) Invt | (6) Invt |
| Treat×Post | 0.0017 | 0.0084*** | 0.0017 | 0.0064** | 0.0068* | 0.0004 |
| | (0.0039) | (0.0028) | (0.0030) | (0.0028) | (0.0039) | (0.0034) |
| Control variables | YES | YES | YES | YES | YES | YES |
| Individual fixed effect | YES | YES | YES | YES | YES | YES |
| Time fixed effect | YES | YES | YES | YES | YES | YES |
| Industry-level clustering | YES | YES | YES | YES | YES | YES |
| N | 8342 | 10305 | 8151 | 10496 | 8963 | 9684 |
| $R^2$ | 0.0714 | 0.0286 | 0.0439 | 0.0463 | 0.0407 | 0.0499 |
| F | 503.5 | 155.3 | 1547 | 372.9 | 10768 | 1443 |

Note

*, **, and *** indicate significant at the 10%, 5%, and 1% levels, respectively, and the heteroscedasticity robust standard errors are in parentheses.

reputation, and stronger financing capabilities. Therefore, the tax incentives brought about by the policy will not bring obvious financing effects to it. Policies will significantly inhibit the investment efficiency of non-state-owned enterprises, while the impact on state-owned enterprises will be insignificant. The possible reason is that state-owned enterprises have hidden policy protection, are large in scale, have low operating risks, have a strong willingness to be provided loans by banks, and have a strong ability to finance themselves. Therefore, the tax incentives brought about by the policy will not significantly improve its financing capacity. The policy will significantly inhibit the investment efficiency of asset-heavy enterprises, while the impact on asset-light enterprises will be insignificant. The possible reason is that asset-heavy companies generally need large-scale fixed investment for manufacturing and face greater financing constraints, and the tax incentives brought about by policies can bring short-term financing to enterprises, alleviate financing constraints, and encourage enterprises to upgrade equipment, expand fixed investment.

However, the manufacturing market is fiercely competitive, profit margins have been shrinking, and investment returns will also decrease. In addition, because fixed investment often has a long return period, companies are forced to short-term loans and long-term investment due to financing pressure, which increases operating risks and increases investment costs. Therefore, although the investment scale of heavy asset enterprises has increased, it is difficult to improve investment efficiency.

## Conclusion

This paper takes China's listed companies from 2000 to 2019 as the research object, and uses the Propensity Score Matching- Difference in Differences model (PSM-DID model) to test the impact of the accelerated depreciation policy of fixed assets in 2014 on the investment efficiency of enterprises. In the theoretical analysis part, it discusses that the policy has both positive and negative effects on the investment efficiency of supported enterprises. In the empirical test part, this policy is used as an exogenous impact, and the PSM-DID model is constructed to identify the causal effect. This article draws the following basic conclusions:

First, the accelerated depreciation policy of fixed assets in 2014 significantly inhibited the investment efficiency of supported enterprises, and this policy was mainly prone to cause excessive investment by enterprises, which in turn caused the loss of investment efficiency, but did not cause obvious underinvestment problems. In terms of dynamic effects, the policy's inhibitory effect on the investment efficiency of enterprises does not have obvious continuous characteristics, and only maintains a relatively large intensity in the year of policy implementation and the first year after implementation. And the inhibitory effect is no longer significant in the second year and later years.

Second, the accelerated depreciation policy of fixed assets in 2014 affected the investment efficiency of supported companies through the financing constraint mechanism. This policy allows companies to accelerate the depreciation of fixed assets in the early period, reduce corporate tax burdens and ease financing constraints by means of tax deferral. In terms of investment efficiency, reducing financing pressure can bring both positive and negative effects, but the overall performance is that the policy suppresses the improvement of corporate investment efficiency by alleviating financing constraints.

Third, in 2014, the accelerated depreciation policy of fixed assets has obvious heterogeneous characteristics in restraining the investment efficiency of enterprises. Specifically, the policy significantly inhibits the investment efficiency of small and medium-sized enterprises, non-state-owned enterprises, and asset-heavy enterprises, while it has no significant impact on large-scale enterprises, state-owned enterprises, and asset-light enterprises.

The weakness of this study is that only one mechanism of financing constraints is identified for the mechanism of the accelerated depreciation policy of fixed assets affecting enterprise investment efficiency, and it is not subdivided according to financing sources. Obviously, the impact mechanism of industrial policy on enterprise behavior is complex, and there are multiple channels of action, and a single action mechanism may not be enough to fully describe the action mechanism of the policy.

The outlook for future research is to look forward to a more in-depth analysis of the mechanism by which the accelerated depreciation policy of fixed assets affects the investment efficiency of enterprises, and to accurately identify more possible mechanisms, such as technological innovation and so on. It can help enterprises to obtain more benefits from the implementation of this policy, which is more conducive to the high-quality development of enterprises.

In general, the current problem faced by Chinese investment is that the scale is large but the efficiency is too low, which also severely limits the high-quality development of the Chinese economy. Investment has always been one of the troikas driving China's economic growth, so the government has continued to introduce various policies to stimulate it. The results of this paper show that policies that stimulate investment do not necessarily improve investment efficiency at the same time, or even inhibit the improvement of investment efficiency. The accelerated depreciation policy of event fixed assets selected in this paper has a very clear boundary between supported industries and unsupported industries, and it is easy to distinguish affected industries from unaffected industries. Moreover, the buffer period from policy introduction to implementation is short, which can be regarded as an exogenous event for individual enterprises. These all create good preconditions for the application of the difference in differences model. The implementation of this policy has caused efficiency losses to corporate investment, mainly because companies are prone to over-investment under the stimulus of the policy. Enterprises should accelerate R&D investment and technological innovation, and improve the efficiency of resource allocation, in order to give full play to the policy advantages, and enterprises can also obtain actual benefits.

## Enlightenment

In the context of China's further deepening of supply-side structural reforms, it is necessary to give full play to the key role of investment and promote the formation of a strong domestic market. For investment, enterprises should not only pay attention to scale, but also pay attention to efficiency. The research in this thesis mainly brings the following two policy connotations:

First, the research results of this thesis will help to understand the economic effects of tax incentive policies more deeply. The policy of accelerated depreciation of fixed assets is not a direct "tax cut" for enterprises, but to reduce the tax burden of enterprises in the initial stage of investment by changing depreciation methods and adjusting the depreciation period. Under the premise of ensuring that the total tax revenue remains unchanged, enterprises are encouraged to upgrade equipment and technology. Existing research has also confirmed that the accelerated depreciation policy of fixed assets has promoted the fixed investment of supported enterprises, indicating that this policy will indeed increase the enthusiasm of enterprises to invest. Different from the existing research, this paper discusses the impact of the accelerated depreciation policy of fixed assets on corporate investment from the perspective of investment efficiency. In terms of investment efficiency, the incentive effect of this policy is not obvious, and even shows a restraining effect. The main reason for this phenomenon is that the improvement of enterprise resource allocation lags behind the growth of investment scale, the production potential of new fixed investment cannot be fully explored, and the technology spillover effect is not obvious. Therefore, enterprises should focus on cultivating high-skilled talents, expanding human capital investment, enhancing R&D and innovation capabilities, improving the matching mechanism between human capital and fixed investment, reducing the negative effects of the policy, and improving investment efficiency.

Second, the formulation and implementation of accelerated depreciation policies for fixed assets should reflect differentiation. The research results of this paper show that the impact of the policy on the investment efficiency of enterprises shows obvious heterogeneity among different enterprise samples, and the investment efficiency losses of small and medium-sized enterprises, non-state-owned enterprises and asset-heavy enterprises are even greater. The design of the policy only distinguishes industries, but ignores the differences between different companies in the same industry. In view of the existence of enterprise heterogeneity, the government should further design accelerated depreciation policies for different enterprise scales, different enterprise ownership properties, and different enterprise asset structures. In the special period of the "new normal", China's economy is facing problems such as overcapacity and declining investment growth, and the accelerated depreciation policy of fixed assets can help achieve the economic goal of "stabilizing investment". What cannot be ignored is that the government must continue to improve the business environment for enterprises, and while expanding domestic demand, it must pay more attention to reasonable and effective investment. Small and medium-sized enterprises, non-state-owned enterprises, and asset-heavy enterprises generally suffer from insufficient financing, which should have been the biggest beneficiaries of the accelerated depreciation policy of fixed assets. However, problems such as insufficient independent innovation and unreasonable resource allocation have become more prominent, making it difficult for new investment to be transformed into actual and effective investment. In the process of policy implementation, the government should pay close attention to the investment changes of these three types of enterprises, and organize relevant technical experts to regularly evaluate the utilization efficiency of their fixed assets. At the same time, pay attention to the supporting use of other related policies and this policy, such as R&D innovation policy, talent introduction policy, etc. Combining production, teaching and research to

increase effective investment and improve investment efficiency. Give full play to the incentive effect of the accelerated depreciation policy of fixed assets to promote the high-quality development of enterprises.

## Supporting information

**S1 Dataset. Benchmark data.**
(XLSX)

**S2 Dataset. Fig 2 Trends in the degree of investment inefficiency.**
(XLSX)

**S3 Dataset. The change trend of the degree of investment inefficiency in the experimental group and the control group.**
(XLSX)

**S4 Dataset. Fig 5 Parallel trend test and dynamic effect test.**
(XLSX)

**S5 Dataset. Fig 6 Nonparametric stochastic simulation.**
(XLSX)

**S6 Dataset. Test after deletion of expanded pilot scope in 2015.**
(XLSX)

**S7 Dataset. Test for excluding the confounding effects of other policies (excluding the three northeastern provinces).**
(XLSX)

**S8 Dataset. Test for Expected effect (excluding 2013).**
(XLSX)

## Author Contributions

**Data curation:** Liangliang Zhai.

**Formal analysis:** Liangliang Zhai.

**Funding acquisition:** Yujing Feng.

**Methodology:** Yujing Feng.

**Resources:** Liping Zhai.

**Supervision:** Fumin Li.

**Visualization:** Liangliang Zhai, Fumin Li.

**Writing – original draft:** Yujing Feng.

**Writing – review & editing:** Liangliang Zhai, Liping Zhai.

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
