## [Decision Letter · Decision Letter 0]

10 May 2022

PONE-D-22-10870Tax Preference, Financing Constraints and Enterprise Investment Efficiency——Experience of China’s Enterprises InvestmentPLOS ONE

Dear Dr. Feng,

Thank you for submitting your manuscript to PLOS ONE. After careful consideration, we feel that it has merit but does not fully meet PLOS ONE’s publication criteria as it currently stands. Therefore, we invite you to submit a revised version of the manuscript that addresses the points raised during the review process.

We look forward to receiving your revised manuscript.

Kind regards,

Ali Safaa Sadiq

Academic Editor

PLOS ONE

Journal Requirements:

5. Please ensure that you refer to Figure 1 in your text as, if accepted, production will need this reference to link the reader to the figure.

Reviewers' comments:

Reviewer's Responses to Questions

**Comments to the Author**

1. Is the manuscript technically sound, and do the data support the conclusions?

Reviewer #1: Yes

Reviewer #2: Yes

2. Has the statistical analysis been performed appropriately and rigorously? 

Reviewer #1: Yes

Reviewer #2: Yes

3. Have the authors made all data underlying the findings in their manuscript fully available?

Reviewer #1: Yes

Reviewer #2: Yes

4. Is the manuscript presented in an intelligible fashion and written in standard English?

Reviewer #1: Yes

Reviewer #2: Yes

5. Review Comments to the Author

Reviewer #1: The manuscript entitled “Tax Preference, Financing 1 Constraints and Enterprise Investment Efficiency-Experience of Chinese Enterprises’ Investment Behavior” is interesting and well written.

This paper takes the 2014 pilot project of accelerated depreciation of fixed assets as a quasi-natural experiment and builds a PSM-DID model based on the data of Chinese listed companies from 2000 to 2019 to test the impact of tax preference on enterprise investment efficiency and its mechanism. The scope of the article is relevant to the journal. Although it is an interesting research article, it required a major revision before a decision to publish it in the journal.

The main concerns are:

1. Based on findings of the study, you must have to write the main policy at the end of the abstract.

2. Define the term “Financing Constraints”, “Enterprise Investment Efficiency”, and “Investment Behavior” in the section of the introduction.

3. Add one more keyword i.e., “China” to the list of keywords.

4. Please write the main contribution of the article at the end of the introduction.

5. After the paragraph of hypothesis 1 (Line 126-128), it is stated that “In this way, under external intervention, the investment structure of enterprises will change, and the alienation of investment behavior will also cause misallocation of resources, which is not conducive to the improvement of productivity”. However, this statement is vague without a proper reference to study. Therefore, I highly recommend updating the statement with the given studies as

“In this way, under external intervention, the investment structure of enterprises will change, and the alienation of investment behavior will also cause misallocation of resources, which is not conducive to the improvement of productivity [1,2]”.

[1] Understanding farmers’ intention and willingness to install renewable energy technology: A solution to reduce the environmental emissions of agriculture. Volume 309, 118459. Published date. 1 March 2022.

[2] Understanding cognitive and socio-psychological factors determining farmers’ intentions to use improved grassland: Implications of land use policy for sustainable pasture production. Land Use Policy, 102, 105250.

6. Line 196-197, the given statement without reference is vague i.e., “First, use the Logit model to estimate the propensity score”. Therefore, you must have to update it with the given studies as

“First, use the Logit model to estimate the propensity score [3]”.

[3] The public policy of agricultural land allotment to agrarians and its impact on crop productivity in Punjab province of Pakistan. Land Use Policy. Volume 90, 104324. Published Date. January 2020.

7. In Equation 1, have you checked the normality of the error term (ε). If not, I highly recommend checking its normality and writing results in the revised paper. Alternatively, you may simply write an assumption with references to given studies as “The given error term (ε) is assumed to be normally distributed with zero mean value and constant variance [4,5].

[4] Application of an artificial neural network to optimise energy inputs: An energy-and cost-saving strategy for commercial poultry farms. Energy. Volume 244, 123169

[5] Extreme weather events risk to crop-production and the adaptation of innovative management strategies to mitigate the risk: A retrospective survey of rural Punjab, Pakistan. Technovation. Volume 117.

8. I highly recommend switching table 1 to the section of empirical results, and therefore, write an explanation of table 1 in the section of empirical results.

9. You must have to write the full forms of variables in the first column of Table 1.

10. Similarly, figure 1, and its explanation must have switched to the section on empirical results.

11. Please write an expansion form of all covariates in the first column of Tables 2 and 3.

12. The equations given in the results must have to shift to the section of methodology.

13. I can see that there are many subsections in the section of results and discussion. I highly recommend writing a separate section for Discussion in the revised article. Therefore, it may reduce the number of subsections of results and discussion.

14. Weaknesses of the study and recommendations for future studies must have to write at the end of the conclusion section.

Reviewer #2: The contribution of this paper is good and I am happy to endorse its acceptance at some point. However, there are several major and minor comments to address. I have listed them as follows:

• In terms of research method and design, there is not much in the paper.

• The comparative algorithms in the experiments are not properly acknowledged and cited

• I also suggest adding some figures to better articular the content as the paper looks very dry at the moment.

• Analysis of the results is missing in the paper. There is a big gap between the results and conclusion. There should be the result analysis between these two sections. After comparing the numerical methods, you have to be able to analyse the results and relate them to their structures. It would be interesting to have your thoughts on why the method works that way? Such analyses would be the core of your work where you prove your understanding of the reason behind the results. You can also link the findings to the hypotheses of the paper. Long story short, this paper requires a very deep analysis from different perspectives

• There is no statistical test to judge about the significance of the numerical method’s results. Without such a statistical test, the conclusion cannot be supported

• There is no discussion on the cost effectiveness of the proposed method. What is the computational complexity? What is the runtime? Please include such discussions. You can also use the big oh notation to show the computation complexity.

• Some mathematical notations and Lemma presentations are not rigorous enough to correctly understand the contents of the paper. The authors are requested to recheck all the definition of variables and further clarify these equations.

6. PLOS authors have the option to publish the peer review history of their article (what does this mean?). If published, this will include your full peer review and any attached files.

Reviewer #1: **Yes: **Ehsan Elahi

Reviewer #2: No

---

## [Author Response · Author response to Decision Letter 0]

14 Jun 2022

Response to editor:

Editor, Journal Requirements# 1: Please ensure that your manuscript meets PLOS ONE's style requirements, including those for file naming. The PLOS ONE style templates can be found at https://journals.plos.org/plosone/s/file?id=wjVg/PLOSOne_formatting_sample_main_body.pdf and https://journals.plos.org/plosone/s/file?id=ba62/PLOSOne_formatting_sample_title_authors_affiliations.pdf

Author response: Our manuscript was revised in accordance with PLOS ONE's style requirements. 

Editor, Journal Requirements# 2: We note that the grant information you provided in the ‘Funding Information’ and ‘Financial Disclosure’ sections do not match. 

Author response: 

Funding Information:

The key project of the National Social Science Foundation of China, "Analysis of the Political Economy of 'Reverse Globalization' and Research on China's Countermeasures" (18AGJ001).

The major project of the National Social Science Foundation of China "Research on China's Solutions for Anti-Globalization Trends and the Reconstruction of International Economic and Trade Rules" (17ZDA097)

Anhui Province University Scientific Research Key Project (Humanities and Social Sciences) "Reform of Security Property Rights System, Financing Constraints and Domestic Value-Added Rate of Exports of Manufacturing Enterprises" (SK2021A0224)

Editor, Journal Requirements# 3: We note that you have indicated that data from this study are available upon request. PLOS only allows data to be available upon request if there are legal or ethical restrictions on sharing data publicly. For more information on unacceptable data access restrictions, please see http://journals.plos.org/plosone/s/data-availability#loc-unacceptable-data-access-restrictions. 

Author response: We have prepared and uploaded the basic data used by the paper.

Editor, Journal Requirements# 4: Please amend either the title on the online submission form (via Edit Submission) or the title in the manuscript so that they are identical.

Author response: We take care of consistency issues when submitting manuscripts.

Editor, Journal Requirements# 5: Please ensure that you refer to Figure 1 in your text as, if accepted, production will need this reference to link the reader to the figure.

Author response: We added the words "(see Fig 1)" after the second paragraph of the section of Policy background and research hypothesis.

Response to reviewer 1:

Reviewer # 1, Concern # 1: Based on findings of the study, you must have to write the main policy at the end of the abstract.

Author response: We added the main policy at the end of the abstract:

“Although the accelerated depreciation policy of fixed assets is conducive to expanding the scale of investment, the incentive effect on investment efficiency is not obvious, and even shows a restraining effect. Given the existence of heterogeneity, the design of the policy should not only distinguish industries, but also pay attention to the differences between different enterprises in the same industry. Strengthening R&D innovation and improving the matching mechanism between human capital and fixed investment will help give full play to the incentive effect of this policy.”

Reviewer # 1, Concern # 2: Define the term “Financing Constraints”, “Enterprise Investment Efficiency”, and “Investment Behavior” in the section of the introduction.

Author response: We have added definitions for these three words at the beginning of the Introduction section. The details are as follows.

“At present, China's economic situation is complex, and uncertainty is rising year by year, and the improvement of investment efficiency has become the focus of scholars' research. Investment has always been one of the "troikas" of China's economic growth, especially since the 2008 financial crisis, the Chinese economy has become more and more dependent on investment. Investment behavior is the process of converting assets with certain value such as monetary funds and manpower into capital, and the investment behavior involved in this thesis is the investment of enterprises in fixed assets. When an enterprise makes an investment, its purpose is theoretically to maximize the value of the enterprise, and such an investment is regarded as an efficient investment. However, in reality, the capital market has various problems such as information asymmetry, principal-agent, transaction costs, etc., so there will be inefficient investment behaviors, such as underinvestment and overinvestment.

Investment efficiency refers to the ratio between the effective results obtained by enterprise investment and the amount of input consumed or occupied, that is, the proportional relationship between the income and expenses, output and input of enterprise investment activities. Investment efficiency is a measure of how effectively an enterprise allocates scarce resources to investment projects and converts investment opportunities into actual investment[1]. In the case of low investment efficiency, it is difficult for enterprises to convert a large number of high-return investment opportunities into actual investment[2]. Improving the efficiency of corporate investment and avoiding invalid investment is an effective way for China to further deepen the supply-side structural reform.

Fazarri et al (1988)[3] put forward the financing constraint hypothesis, that is, in an imperfect capital market, information asymmetry, agency problems and related transaction costs make the internal and external financing costs of enterprises different, resulting in their external financing being constrained, which creates a situation that makes the business significantly dependent on internal funding. Under the circumstance of financing constraints, enterprises lack sensitivity to changes in capital costs, asset prices and investment opportunities, thus affecting the improvement of enterprise value. Due to the existence of transaction costs in the capital market, in a sense, all enterprises are faced with a certain degree of financing constraints, which is the resistance faced by enterprises in financing all their feasible investments.”

[1] Xuesong Q, Sheng F. Enactment of the Property Law, Financial Constraints and Investment Efficiency of Private-Owned Enterprises——Empirical Analysis Based on Difference-in-Differences Method. China Economic Quarterly 2021; 21:713-32.

 [2] Kun Y, Zhiguo L, Xiaorong Z, Jiangang X. Investment Efficiency Puzzle: Financial Constraint Hypothesis and Monetary Policy Shock. Economic Research Journal 2014; 49:106-20.

 [3] Fazzari SM, Hubbard RG, Petersen BC. Financing Constraints and Corporate Investment. 1988.

Reviewer # 1, Concern # 3: Add one more keyword i.e., “China” to the list of keywords.

Author response: We have added “China” to the list of keywords.

Reviewer # 1, Concern # 4: Please write the main contribution of the article at the end of the introduction.

Author response: The main contribution of this paper has been stated in the last paragraph of the Introduction section of our original paper. The details are as follows:

“Compared with the existing literature, the marginal contributions of this paper are mainly as follows: First of all, this paper studies the impact of preferential tax policies on enterprise investment from the perspective of investment efficiency, makes up for the relative lack of research in this field, and helps to understand the investment effects of preferential tax policies more comprehensively. Secondly, we use the exogenous impact of the accelerated depreciation policy of fixed assets as a quasi-natural experiment, and construct a Difference in Differences model (DID model) for causal effect identification, which can effectively overcome the endogenous problems in the empirical test, thereby more accurately assessing the effect of tax preferential policies on enterprises investment efficiency. Thirdly, this paper identifies the mediating role of financing constraints and explores its possible mechanism in depth.”

Reviewer # 1, Concern # 5: After the paragraph of hypothesis 1 (Line 126-128), it is stated that “In this way, under external intervention, the investment structure of enterprises will change, and the alienation of investment behavior will also cause misallocation of resources, which is not conducive to the improvement of productivity”. However, this statement is vague without a proper reference to study. Therefore, I highly recommend updating the statement with the given studies as

“In this way, under external intervention, the investment structure of enterprises will change, and the alienation of investment behavior will also cause misallocation of resources, which is not conducive to the improvement of productivity [1,2]”.

[1] Understanding farmers’ intention and willingness to install renewable energy technology: A solution to reduce the environmental emissions of agriculture. Volume 309, 118459. Published date. 1 March 2022.

[2] Understanding cognitive and socio-psychological factors determining farmers’ intentions to use improved grassland: Implications of land use policy for sustainable pasture production. Land Use Policy, 102, 105250.

Author response: We have added the corresponding references you provided as “In this way, under external intervention, the investment structure of enterprises will change, and the alienation of investment behavior will also cause misallocation of resources, which is not conducive to the improvement of productivity [23,24]”.

[23] Understanding farmers’ intention and willingness to install renewable energy technology: A solution to reduce the environmental emissions of agriculture. Volume 309, 118459. Published date. 1 March 2022.

[24] Understanding cognitive and socio-psychological factors determining farmers’ intentions to use improved grassland: Implications of land use policy for sustainable pasture production. Land Use Policy, 102, 105250.

Reviewer # 1, Concern # 6: Line 196-197, the given statement without reference is vague i.e., “First, use the Logit model to estimate the propensity score”. Therefore, you must have to update it with the given studies as “First, use the Logit model to estimate the propensity score [3]”.

[3] The public policy of agricultural land allotment to agrarians and its impact on crop productivity in Punjab province of Pakistan. Land Use Policy. Volume 90, 104324. Published Date. January 2020.

Author response: We have added the corresponding references you provided as “First, use the Logit model to estimate the propensity score [25]”.

[25] The public policy of agricultural land allotment to agrarians and its impact on crop productivity in Punjab province of Pakistan. Land Use Policy. Volume 90, 104324. Published Date. January 2020.

Reviewer # 1, Concern # 7: In equation 1, have you checked the normality of the error term (ε). If not, I highly recommend checking its normality and writing results in the revised paper. Alternatively, you may simply write an assumption with references to given studies as “The given error term (ε) is assumed to be normally distributed with zero mean value and constant variance [4,5].

[4] Application of an artificial neural network to optimise energy inputs: An energy-and cost-saving strategy for commercial poultry farms. Energy. Volume 244, 123169

[5] Extreme weather events risk to crop-production and the adaptation of innovative management strategies to mitigate the risk: A retrospective survey of rural Punjab, Pakistan. Technovation. Volume 117.

Author response: In equation 1, we have added the corresponding references you provided as “The given error term (ε) is assumed to be normally distributed with zero mean value and constant variance [26,27].

[26] Application of an artificial neural network to optimise energy inputs: An energy-and cost-saving strategy for commercial poultry farms. Energy. Volume 244, 123169

[27] Extreme weather events risk to crop-production and the adaptation of innovative management strategies to mitigate the risk: A retrospective survey of rural Punjab, Pakistan. Technovation. Volume 117.

Reviewer # 1, Concern # 8: I highly recommend switching table 1 to the section of empirical results, and therefore, write an explanation of table 1 in the section of empirical results.

Author response: Table 1 is only descriptive statistics of the data used in the empirical study, not the empirical results. Descriptive statistics is a method of summarizing and expressing quantitative data in a way that reveals the distributional properties of the data. It is used to summarize and characterize data, often as the basis for further quantitative analysis of the data, or as an effective complement to inferential statistical methods. In short, it uses a series of complex data to represent the data set with representative numbers through the analysis of the characteristics of the data distribution, reflecting the overall situation of the data set. In summary, we believe that the location of Table 1 is reasonable.

Reviewer # 1, Concern # 9: You must have to write the full forms of variables in the first column of Table 1.

Author response: We added a column to the left of the original Table 1 to show the full names of the variables.

variable Variables abbreviation

the degree of Enterprise investment inefficiency 

the degree of enterprise overinvestment 

the degree of enterprise underinvestment 

Grouped dummy variable 

staged dummy variable 

the size of the enterprise 

company listing time 

corporate internal cash flow 

corporate asset-liability ratio 

the fixed assets ratio 

the growth rate of the total assets of the enterprise 

enterprise asset return 

enterprise book-to-market value ratio 

Reviewer # 1, Concern # 10: Similarly, figure 1, and its explanation must have switched to the section on empirical results.

Author response: Figure 1 shows the mechanism by which the accelerated depreciation policy of fixed assets affects the investment efficiency of enterprises, rather than the empirical results. Therefore, we believe that Figure 1 should be kept in the original text position for the intuitive display of the mechanism of action.

Reviewer # 1, Concern # 11: Please write an expansion form of all covariates in the first column of Tables 2 and 3.

Author response: We added a column to the left of the original Table 2 to show the full names of the covariates. The first two columns of Table 2 are shown below.

Covariates Covariates abbreviation

the size of the enterprise 

company listing time 

corporate internal cash flow 

corporate asset-liability ratio 

the growth rate of the total assets of the enterprise 

enterprise book-to-market value ratio 

enterprise asset return 

the fixed assets ratio 

We added the full name of the variable in front of the corresponding variable abbreviation in Table 3. The revised Table 3 is shown below.

Table 3 Benchmark regression results

Variables (1) (2) (3) (4) (5) (6) (7) (8)

 the degree of Enterprise investment inefficiency

( )

the degree of Enterprise investment inefficiency

( )

the degree of Enterprise investment inefficiency

( )

the degree of Enterprise investment inefficiency

( )

the degree of Enterprise investment inefficiency

( )

the degree of Enterprise investment inefficiency

( )

the degree of Enterprise investment inefficiency

( )

the degree of Enterprise investment inefficiency

( )

Interaction item

( )

0.0053** 0.0049** -0.0051*** 0.0040** 0.0037* 0.0050** 0.0053*** 0.0049**

 (0.0020) (0.0023) (0.0014) (0.0016) (0.0019) (0.0022) (0.0020) (0.0024)

the size of the enterprise ( 

 0.0122*** 0.0123*** 0.0061*** 0.0122***

 (0.0020) (0.0021) (0.0012) (0.0016)

company listing time ( )

 -0.0027*** -0.0028*** -0.0018*** -0.0027***

 (0.0003) (0.0003) (0.0001) (0.0003)

corporate internal cash flow ( )

 -0.0002 -0.0039 -0.0003 -0.0002

 (0.0084) (0.0102) (0.0077) (0.0078)

corporate asset-liability ratio ( )

 0.0153*** 0.0151*** 0.0228*** 0.0153***

 (0.0051) (0.0047) (0.0037) (0.0042)

the fixed assets ratio ( )

 0.0761** 0.0795*** 0.0825*** 0.0761***

 (0.0284) (0.0275) (0.0244) (0.0215)

enterprise book-to-market value ratio 

 -0.0180*** -0.0203*** -0.0160*** -0.0180***

 (0.0042) (0.0033) (0.0038) (0.0038)

enterprise asset return ( )

 0.0140** 0.0148** 0.0141** 0.0140*

 (0.0066) (0.0064) (0.0059) (0.0075)

the growth rate of the total assets of the enterprise ( )

 0.0003 0.0003 0.0004** 0.0003

 (0.0002) (0.0002) (0.0002) (0.0004)

Individual fixed effect YES YES YES YES NO NO YES YES

Time fixed effect YES YES NO NO YES YES YES YES

Industry-level clustering YES YES YES YES YES YES NO NO

N 18647 18647 18647 18647 18647 18647 18647 18647

R2 0.0262 0.0385 0.0009 0.0310 0.0256 0.0354 0.0262 0.0385

Note: *, **, and *** indicate significant at the 10%, 5%, and 1% levels, respectively, and the heteroscedasticity robust standard errors are in parentheses.

Reviewer # 1, Concern # 12: The equations given in the results must have to shift to the section of methodology.

Author response: These equations are built to test the robustness of the empirical results, not the benchmark regression at the heart of this paper. Putting them together will make the content of the article less contextual and cluttered.

Reviewer # 1, Concern # 13: I can see that there are many subsections in the section of results and discussion. I highly recommend writing a separate section for Discussion in the revised article. Therefore, it may reduce the number of subsections of results and discussion.

Author response: We consider the existence of these subsections to be necessary as they further examine the robustness, dynamics, mechanism of action, and heterogeneity of the empirical results. And these detailed tests make the empirical results more reliable.

Reviewer # 1, Concern # 14: Weaknesses of the study and recommendations for future studies must have to write at the end of the conclusion section.

Author response: We accepted your comments and added weaknesses in this study and an outlook for future research at the end of the conclusion.

The weakness of this study is that only one mechanism of financing constraints is identified for the mechanism of the accelerated depreciation policy of fixed assets affecting enterprise investment efficiency, and it is not subdivided according to financing sources. We know that the impact mechanism of industrial policy on enterprise behavior is complex, and there are multiple channels of action, and a single action mechanism may not be enough to fully describe the action mechanism of the policy.

Our outlook for future research is to look forward to a more in-depth analysis of the mechanism by which the accelerated depreciation policy of fixed assets affects the investment efficiency of enterprises, and to accurately identify more possible mechanisms, such as technological innovation and so on. It can help enterprises to obtain more benefits from the implementation of this policy, which is more conducive to the high-quality development of enterprises.

Response to reviewer 2:

Reviewer # 2, Concern # 1: In terms of research method and design, there is not much in the paper.

Author response: We accept your comments and have appropriately supplemented the research methodology and design used at the end of the first paragraph of the appropriate sections (Model setting and variable selection) of the original paper. The added paragraphs are as follows.

“Difference-in-difference (DID) has been a very popular method in recent years to assess the effects of regional policies. The basic idea is to take the regional policy as a quasi-natural experiment, differentiate the experimental group under the influence of the policy and the control group not affected by the policy before and after the implementation of the regional policy, and then calculate the difference between the two groups of difference results, so as to obtain the net regional effect of the policy. However, a reasonable assessment of regional policy should first ensure that both the experimental and control groups are randomly selected, thereby avoiding the self-selection problem. In fact, the division of the experimental group and the control group is often not randomly selected, and there are different characteristics, which will cause the selectivity bias of the difference-in-difference method and further lead to endogeneity problems. Since the introduction of the accelerated depreciation policy for fixed assets has a great impact on heavy-asset enterprises, it is obviously not a random selection, which may lead to the problem of sample selection bias. Furthermore, to reasonably evaluate a regional policy, there needs to be a suitable control group, i.e., first, the experimental group and the control group are similar, and the experimental group is affected by the policy and the control group is not affected by the policy.

The propensity score matching method (PSM) is usually used to solve the problem of selection bias, and its basic idea is to form an approximate randomized experiment by constructing a counterfactual framework. The so-called counterfactual refers to observing the consequences of the experimental group without policy intervention through the control group, and then comparing the two results to eliminate the problem of selection bias, so as to obtain the true causal relationship. The PSM-DID method is to first use the PSM method to eliminate the selection bias in the sample, and then use the DID method to identify the causal effect.”

Reviewer # 2, Concern # 2: The comparative algorithms in the experiments are not properly acknowledged and cited. 

Author response: On the basis of your first comment, we have added relevant method acknowledgments and citations and listed the references below.

“The PSM-DID method has been widely recognized and applied. For example, Wang Zhiyong (2022)[25] used the PSM-DID method to evaluate the industrial efficiency of the revitalization policy of old industrial bases in Northeast China. Gong Maogang and Zhang Meijiao (2022)[26] used the PSM-DID method to study the positive impact of the "three rights separation" of contracted land and agricultural subsidies on agricultural mechanization. In addition, a large number of scholars such as Zhang Minglin and Li Huaxu (2021)[27] and Si Chunxiao (2021)[28] have applied the PSM-DID method to academic research, and obtained scientific and reasonable conclusions.”

[25] Zhiyong W. Financial Resources, Industrial Efficiency and Issues of Northeast China——An Empirical Analysis Based on Prefectural Panel Data During 2001-2016. Modern Economic Science 2022; 44:104-18.

[26] Maogang G, Meijiao Z. Research on the Effect of “Separation of Three Rights” of Contracted Land and Agricultural Subsidies on Agricultural Mechanization: An Empirical Analysis Based on PSM-DID Method. Statistical Research:1-16.

[27] Minglin Z, Huaxu L. Effect Evaluation of State Priority Support Policies on Promoting Green TFP: Empirical Evidence from Old Revolutionary Areas. Journal of Finance and Economics 2021; 47:65-79.

[28] Chunxiao S, Shiyi S, Changyuan L. The Impact of Free Trade Zone on FDI Inflows: Evidence Based on PSM-DID. World Economy Studies 2021:9-23.

Reviewer # 2, Concern # 3: I also suggest adding some figures to better articular the content as the paper looks very dry at the moment.

Author response: In order to make the content of the article interesting and attractive enough, many figures and tables have been inserted in this article. 

For example, Figure 2, which describes the trend of investment inefficiency of listed companies over time, includes the degree of over-investment and the degree of under-investment. On the whole, the degree of investment inefficiency shows a downward trend, that is, the investment efficiency is increasing. However, the decline in investment inefficiency is very slow. From the curve of the sample period 2000-2019 in this paper, there are two obvious fluctuations, the earlier one is the increase in inefficiency in 2007, and the second is the increase in inefficiency in 2014-2015. The policy object studied in this paper is the accelerated depreciation policy of fixed assets implemented in 2014, which coincides with the second rising period of investment inefficiency. Accordingly, a bold conjecture arises: whether the implementation of the accelerated depreciation policy of fixed assets will increase the degree of investment inefficiency, which is also the question that this paper attempts to verify. Judging from the research sample as a whole, Chinese enterprises have both over-investment and under-investment problems in investment, and over-investment is more serious than under-investment. Overinvestment and underinvestment followed the same trend as overall investment inefficiency, with one of the most striking differences being that during 2014-2015, overinvestment increased sharply, while underinvestment increased very gently. Does this mean that if the accelerated depreciation policy of fixed assets does lead to inefficiency of investment, whether it is possible that the policy leads to over-investment, and then over-investment leads to the loss of investment efficiency, this will be another question to be discussed in this paper.

For example, Table 1 shows the descriptive statistics of the main variables, including the observer, mean, standard deviation, minimum and maximum values of each variable. There are 18,647 observations in the research sample in this paper, including 6,709 over-investment observations and 11,938 under-investment observations. The average value of the enterprise's investment inefficiency Invt is 0.0445, the minimum value is 0.00001, and the maximum value is 0.4717, which shows that there are large differences between different enterprises. The average value of Over_Invt of corporate overinvestment degree is 0.0618, the minimum value is 0.000015, and the maximum value is 0.4717; the average value of corporate underinvestment degree Under_Invt is 0.0348, the minimum value is 0.00001, and the maximum value is 0.2657. The mean value of the grouping dummy variable Treat is 0.4082, indicating that nearly 41% of the companies in the research sample belong to six major industries, which will be affected by the accelerated depreciation policy of fixed assets, and the selection of the experimental group is more representative. The mean value of the staging dummy variable Post is 0.5160, indicating that the period after the implementation of the policy accounts for more than 51% of the entire sample period, and the selection of the sample period has a good representativeness. Other variables in the model are also listed in Table 1. 

In addition, in the robustness test and placebo test of the empirical results, there are also corresponding figures and tables to explain the results.

Reviewer # 2, Concern # 4: Analysis of the results is missing in the paper. There is a big gap between the results and conclusion. There should be the result analysis between these two sections. After comparing the numerical methods, you have to be able to analyse the results and relate them to their structures. It would be interesting to have your thoughts on why the method works that way? Such analyses would be the core of your work where you prove your understanding of the reason behind the results. You can also link the findings to the hypotheses of the paper. Long story short, this paper requires a very deep analysis from different perspectives

Author response: Modifications have been made in accordance with comments. We discuss the reasons behind the empirical findings in the context of current era themes and policy contexts. Table 3 in the paper shows the benchmark regression results of this paper, and the first paragraph of the Benchmark inspection section has explained the sign and significance of the coefficients of the core explanatory variables, and linked the empirical results with the research hypothesis proposed above. And the second and third paragraph of the Benchmark inspection section has explained the situation of other control variables. 

After a detailed explanation of the benchmark regression results, the paper makes appropriate extensions on the basis of the results of this study. Combined with contemporary themes and policy contexts, try to understand the deeper reasons behind the findings. The added paragraph at the end of the Benchmark inspection subsection reads:

“The prosperity of China's economy is largely driven by investment, but the success of investment depends not only on size, but also on efficiency. Investment efficiency refers to the proportional relationship between the income obtained by an enterprise's investment and the resources it consumes or occupies. It is true that the implementation of the accelerated depreciation policy for fixed assets has achieved obvious effects, such as easing the financing constraints of enterprises, reducing leverage, and encouraging enterprise growth (Liu Xing et al., 2019[10]; Tong Jinzhi et al., 2020[37]; Zhong Guohui et al., 2021[38]), and improving the cash flow of enterprises will promote the growth of investment scale (Xiong Bo and Du Jiaqi, 2020)[24]. This paper also finds that the accelerated depreciation policy of fixed assets will inhibit the improvement of investment efficiency of supported enterprises. After analyzing the positive and negative effects of the policy on the investment efficiency of enterprises, the empirical test shows that the negative effects will offset the positive effects, and the policy will have an adverse impact on the investment efficiency of supported enterprises as a whole. At present, China's economy is in a historical stage of further deepening supply-side structural reform and strengthening demand-side management. It is necessary to pay more attention to the improvement of investment efficiency while paying attention to the increase in the scale of enterprise investment. Looking at the world, the forces of anti-globalization are intensifying, and the new coronavirus epidemic is raging around the world, and the investment of enterprises is inevitably affected. For enterprises, although the accelerated depreciation policy for fixed assets does not change the total amount of depreciation and tax deduction in the useful life, it allows enterprises to accrue a large amount of depreciation in the current period when they purchase fixed assets, which can be regarded as a tax preference for deferred taxation. This policy releases the vitality of enterprises by reducing the current tax burden and easing financing constraints, such as promoting R&D investment (Li Haoyang et al., 2017)[13] and improving the structure of human capital (Liu Qiren and Zhao Can, 2020)[14]. It is undeniable that the policy will also have negative effects, such as inhibiting the improvement of the investment efficiency of supported enterprises, which is related to the effectiveness of enterprise investment, and will also affect other characteristics and behaviors of enterprises, and even offset the policy's effect to a certain extent. positive influence. Combined with the domestic and international situation, it is necessary to comprehensively analyze the advantages and disadvantages of the accelerated depreciation policy of fixed assets for enterprises, so as to maximize the strengths and avoid weaknesses, and fully tap the incentive effect of the policy, which will help revitalize the market and promote the high-quality development of enterprises.”

Reviewer # 2, Concern # 5: There is no statistical test to judge about the significance of the numerical method’s results. Without such a statistical test, the conclusion cannot be supported

Author response: We accept comments, which have been revised in the original text. According to the comments to be revised, the results of the F test and Wald test of the regression results have been supplemented in the original tables.

Reviewer # 2, Concern # 6: There is no discussion on the cost effectiveness of the proposed method. What is the computational complexity? What is the runtime? Please include such discussions. You can also use the big oh notation to show the computation complexity.

Author response: We accept comments and have made adjustments in the original paper. According to the comments that need to be revised, in the Model setting and variable selection section of the original paper, the research methods and design used in the paper are clarified, and we have added the advantages of the PSM-DID method compared to the DID method. An addition to this section was included in response to your first comment, so the details will not be repeated here.

Reviewer # 2, Concern # 7: Some mathematical notations and Lemma presentations are not rigorous enough to correctly understand the contents of the paper. The authors are requested to recheck all the definition of variables and further clarify these equations.

Author response: We have made revisions in the original paper based on comments. According to the comments to be revised, the definition of each variable has been supplemented in Table 1, which makes the content of the paper and the measurement model easier to understand.

Table 1 Descriptive statistics of main variables

variable Variables abbreviation Variables definition Observations Average Standard deviation Minimum Maximum

the degree of Enterprise investment inefficiency The degree of inefficiency of business investment 18647 0.0445 0.0524 0.00001 0.4717

the degree of enterprise overinvestment Excessive degree of corporate investment 6709 0.0618 0.0765 0.000015 0.4717

the degree of enterprise underinvestment Insufficient level of business investment 11938 0.0348 0.0272 0.00001 0.2657

Grouped dummy variable Grouping dummy variable, which indicates whether it belongs to the industry supported by the accelerated depreciation policy of fixed assets, if it belongs to the industry, the value is 1, otherwise the value is 0 18647 0.4082 0.4915 0 1

staged dummy variable Stage dummy variable, which indicates whether it is after the implementation year of the accelerated depreciation policy for fixed assets, if so, the value is 1, otherwise the value is 0 18647 0.5160 0.4998 0 1

the size of the enterprise Enterprise size, which is the natural logarithm of total assets at the end of the year. 18647 21.9157 1.1637 17.1219 27.4677

company listing time The number of years the company has been listed, which is calculated by subtracting the listing year from the statistical year. 18647 9.1892 5.9331 2 29

corporate internal cash flow Internal business cash flow, which is calculated by dividing net cash flow from operating activities at the end of the year by total assets. 18647 0.0486 0.0738 -1.9377 0.4876

corporate asset-liability ratio The company's asset-liability ratio, which is calculated by dividing total liabilities by total assets at the end of the year. 18647 0.3724 0.2268 0 1.1315

the fixed assets ratio The firm's fixed asset ratio, which is calculated as year-end fixed assets divided by total assets. 18647 0.0436 0.0417 0 0.6773

the growth rate of the total assets of the enterprise The growth rate of the company's total assets. It is calculated by subtracting the total assets at the end of the previous year from the total assets at the end of the current year, and then dividing the total assets at the end of the previous year. 18647 0.0106 1.0878 -66.5353 0.9900

enterprise asset return Corporate return on assets, which is calculated by dividing year-end net profit by total assets. 18647 0.0306 0.1142 -8.7534 0.3999

enterprise book-to-market value ratio The company's book-to-market ratio, which is calculated by dividing total assets by market value at the end of the year. 18647 0.6149 0.2560 0 6.5459

All modifications were made using the MS Word Track Changes function.

---

## [Decision Letter · Decision Letter 1]

10 Jul 2022

PONE-D-22-10870R1Tax Preference, Financing Constraints and Enterprise Investment Efficiency——Experience of China’s Enterprises InvestmentPLOS ONE

Dear Dr. Feng,

Thank you for submitting your manuscript to PLOS ONE. After careful consideration, we feel that it has merit but does not fully meet PLOS ONE’s publication criteria as it currently stands. Therefore, we invite you to submit a revised version of the manuscript that addresses the points raised during the review process.

We look forward to receiving your revised manuscript.

Kind regards,

Ali Safaa Sadiq

Academic Editor

PLOS ONE

Journal Requirements:

Please review your reference list to ensure that it is complete and correct. If you have cited papers that have been retracted, please include the rationale for doing so in the manuscript text, or remove these references and replace them with relevant current references. Any changes to the reference list should be mentioned in the rebuttal letter that accompanies your revised manuscript. If you need to cite a retracted article, indicate the article’s retracted status in the Reference.

Reviewers' comments:

Reviewer's Responses to Questions

**Comments to the Author**

1. If the authors have adequately addressed your comments raised in a previous round of review and you feel that this manuscript is now acceptable for publication, you may indicate that here to bypass the “Comments to the Author” section, enter your conflict of interest statement in the “Confidential to Editor” section, and submit your "Accept" recommendation.

Reviewer #1: All comments have been addressed

Reviewer #2: (No Response)

2. Is the manuscript technically sound, and do the data support the conclusions?

Reviewer #1: Yes

Reviewer #2: (No Response)

3. Has the statistical analysis been performed appropriately and rigorously? 

Reviewer #1: Yes

Reviewer #2: (No Response)

4. Have the authors made all data underlying the findings in their manuscript fully available?

Reviewer #1: Yes

Reviewer #2: (No Response)

5. Is the manuscript presented in an intelligible fashion and written in standard English?

Reviewer #1: Yes

Reviewer #2: (No Response)

6. Review Comments to the Author

Reviewer #1: (No Response)

Reviewer #2: Some final cosmetic comments:

* The results of your comparative study should be discussed in-depth and with more insightful comments on the behaviour of your algorithm on various case studies. Discussing results should not mean reading out the tables and figures once again.

* Avoid lumping references as in [x, y] and all other. Instead summarize the main contribution of each referenced paper in a separate sentence. For scientific and research papers, it is not necessary to give several references that say exactly the same. Anyway, that would be strange, since then what is innovative scientific contribution of referenced papers? For each thesis state only one reference.

* Avoid using first person.

* Avoid using abbreviations and acronyms in title, abstract, headings and highlights.

* Please avoid having heading after heading with nothing in between, either merge your headings or provide a small paragraph in between.

* The first time you use an acronym in the text, please write the full name and the acronym in parenthesis. Do not use acronyms in the title, abstract, chapter headings and highlights.

* The results should be further elaborated to show how they could be used for the real applications.

7. PLOS authors have the option to publish the peer review history of their article (what does this mean?). If published, this will include your full peer review and any attached files.

Reviewer #1: **Yes: **Ehsan Elahi

Reviewer #2: No

---

## [Author Response · Author response to Decision Letter 1]

26 Jul 2022

Response to editor:

Editor, Journal Requirements# 1: Please review your reference list to ensure that it is complete and correct. If you have cited papers that have been retracted, please include the rationale for doing so in the manuscript text, or remove these references and replace them with relevant current references. Any changes to the reference list should be mentioned in the rebuttal letter that accompanies your revised manuscript. If you need to cite a retracted article, indicate the article’s retracted status in the Reference.

Author response: We checked each paper in the reference list individually and found that the information in two papers was inaccurate.

1. The third reference is a working paper. In the original manuscript, the year was mistakenly written as 1988, which has now been changed to the correct year of 1987.

2. The year is omitted in the 29th reference, which has been supplemented.

Other than that, no retracted references were found. If editors or experts find such a situation, please let us know in detail.

Author, Request for Modification of Funding Information: 

Author: 

Our Funding Information has been changed to remove "The major project of the National Social Science Foundation of China "Research on China's Solutions for Anti-Globalization Trends and the Reconstruction of International Economic and Trade Rules" (17ZDA097)", and added "Youth Project of Natural Science Foundation of Anhui Province, China. "Economic Policy Uncertainty and Servicization of Manufacturing Enterprises: Theoretical Framework, Impact and Identification of Causal Effects" (2208085QG222)". So the new Funding Information list is as follows:

Funding Information:

The key project of the National Social Science Foundation of China, "Analysis of the Political Economy of 'Reverse Globalization' and Research on China's Countermeasures" (18AGJ001).

Anhui Province University Scientific Research Key Project (Humanities and Social Sciences) "Reform of Security Property Rights System, Financing Constraints and Domestic Value-Added Rate of Exports of Manufacturing Enterprises" (SK2021A0224)

Youth Project of Natural Science Foundation of Anhui Province, China. "Economic Policy Uncertainty and Servicization of Manufacturing Enterprises: Theoretical Framework, Impact and Identification of Causal Effects" (2208085QG222)

Response to reviewer 2:

Reviewer # 2, Concern # 1: The results of your comparative study should be discussed in-depth and with more insightful comments on the behaviour of your algorithm on various case studies. Discussing results should not mean reading out the tables and figures once again.

Author response: We have conducted a more in-depth analysis of the empirical results in the second paragraph of the Further expand the analysis section of the original manuscript, and for further depth, we have added the following.

“This policy aims to reduce the income tax burden in the early stage of investment, promote enterprises to speed up the renewal of machinery and equipment, and stimulate research and development innovation. The emergence of over-investment may be due to the low efficiency of resource allocation within the enterprise, or even the existence of resource misallocation, and the lag in the upgrading of human capital structure, which cannot match suitable human capital for newly increased fixed asset investment. Therefore, the investment made by the enterprise under the incentive of the policy can easily become the repeated investment, the production potential cannot be fully tapped, the equipment upgrade cannot produce the technology spillover effect, and the productivity and R&D innovation of the enterprise cannot be improved accordingly. ”

Reviewer # 2, Concern # 2: Avoid lumping references as in [x, y] and all other. Instead summarize the main contribution of each referenced paper in a separate sentence. For scientific and research papers, it is not necessary to give several references that say exactly the same. Anyway, that would be strange, since then what is innovative scientific contribution of referenced papers? For each thesis state only one reference.

Author response: In order to understand our research on this topic in as much detail as possible, we have read a large number of references, and it is very common that many literatures have been studied from different angles or methods and have reached the same point of view or conclusion. So it is unavoidable to have multiple reference marks after the same idea. We also checked recent papers published in the journal of PLOS ONE, many of which have a similar situation. Therefore we have not made changes here.

Reviewer # 2, Concern # 3: Avoid using first person.

Author response: We accept comments, and have revised all first-person descriptions.

Reviewer # 2, Concern # 4: Avoid using abbreviations and acronyms in title, abstract, headings and highlights.

Author response: We have supplemented the abbreviations in the abstract with the full name. We have paid attention to this in the process of writing the paper, and the full name of abbreviations will be displayed for the first time in the paper. However, if you write the full name directly in the title of the chapter, it will cause the title to be too long and affect the appearance. But we are sure that before the title appears abbreviations, the full name of the abbreviations has already been explained in the previous text.

Reviewer # 2, Concern # 5: Please avoid having heading after heading with nothing in between, either merge your headings or provide a small paragraph in between.

Author response: In order to avoid this situation, we moved Table 2 to the bottom of the analysis content to avoid the connection between the header and the title. The titles of other individual chapters and section titles are next to each other, referring to the structure of articles published in the journal of PLOS ONE.

Reviewer # 2, Concern # 6: The first time you use an acronym in the text, please write the full name and the acronym in parenthesis. Do not use acronyms in the title, abstract, chapter headings and highlights.

Author response: We accept comments and have made adjustments in the original paper. We have supplemented the full name and the acronym in parenthesis. For example, the full names of PSM-DID and RD are supplemented in the abstract. Added the full name of ST that appears in the last paragraph of the section Sample selection and data sources.

research and development (R&D) 

Propensity Score Matching – Difference in Differences (PSM-DID)

ST (Special Treatment)

Reviewer # 2, Concern # 7: The results should be further elaborated to show how they could be used for the real applications.

Author response: We added further details at the end of the Conclusion section to show how they can be used in practical applications.

“In general, the current problem faced by Chinese investment is that the scale is large but the efficiency is too low, which also severely limits the high-quality development of the Chinese economy. Investment has always been one of the troikas driving China's economic growth, so the government has continued to introduce various policies to stimulate it. The results of this paper show that policies that stimulate investment do not necessarily improve investment efficiency at the same time, or even inhibit the improvement of investment efficiency. The accelerated depreciation policy of event fixed assets selected in this paper has a very clear boundary between supported industries and unsupported industries, and it is easy to distinguish affected industries from unaffected industries. Moreover, the buffer period from policy introduction to implementation is short, which can be regarded as an exogenous event for individual enterprises. These all create good preconditions for the application of the difference in differences model. The implementation of this policy has caused efficiency losses to corporate investment, mainly because companies are prone to over-investment under the stimulus of the policy. Enterprises should accelerate R&D investment and technological innovation, and improve the efficiency of resource allocation, in order to give full play to the policy advantages, and enterprises can also obtain actual benefits.”

All modifications were made using the MS Word Track Changes function.

---

## [Decision Letter · Decision Letter 2]

3 Aug 2022

PONE-D-22-10870R2Tax Preference, Financing Constraints and Enterprise Investment Efficiency——Experience of China’s Enterprises InvestmentPLOS ONE

Dear Dr. Feng,

Thank you for submitting your manuscript to PLOS ONE. After careful consideration, we feel that it has merit but does not fully meet PLOS ONE’s publication criteria as it currently stands. Therefore, we invite you to submit a revised version of the manuscript that addresses the points raised during the review process.

We look forward to receiving your revised manuscript.

Kind regards,

Ali Safaa Sadiq

Academic Editor

PLOS ONE

Journal Requirements:

Additional Editor Comments (if provided):

The authors have substantially addressed most of the given feedback and comments. Though, there are some left over minor changes given by the second reviewer that need to be addressed before the final acceptance. Hence, I would like to invite the authors for another round of minor changes.

Reviewers' comments:

Reviewer's Responses to Questions

**Comments to the Author**

1. If the authors have adequately addressed your comments raised in a previous round of review and you feel that this manuscript is now acceptable for publication, you may indicate that here to bypass the “Comments to the Author” section, enter your conflict of interest statement in the “Confidential to Editor” section, and submit your "Accept" recommendation.

Reviewer #1: All comments have been addressed

Reviewer #2: (No Response)

2. Is the manuscript technically sound, and do the data support the conclusions?

Reviewer #1: Yes

Reviewer #2: (No Response)

3. Has the statistical analysis been performed appropriately and rigorously? 

Reviewer #1: Yes

Reviewer #2: (No Response)

4. Have the authors made all data underlying the findings in their manuscript fully available?

Reviewer #1: Yes

Reviewer #2: (No Response)

5. Is the manuscript presented in an intelligible fashion and written in standard English?

Reviewer #1: Yes

Reviewer #2: (No Response)

6. Review Comments to the Author

Reviewer #1: The authors have addressed all previous comments and now the paper is ready for publication. The authors have addressed all previous comments and now the paper is ready for publication

Reviewer #2: Excellent effort on the revision. Just one final check:

• Are all the images used in this work copyrights free? If not, have the authors obtained proper copyrights permission to re-use them? Please kindly clarify, and this is just to ensure all the figures are fine to be published in this work.

• Also, the list of references should be carefully checked to ensure consistency with between all references and their compliances with the journal policy on referencing.

7. PLOS authors have the option to publish the peer review history of their article (what does this mean?). If published, this will include your full peer review and any attached files.

Reviewer #1: **Yes: **Ehsan Elahi

Reviewer #2: No

---

## [Author Response · Author response to Decision Letter 2]

7 Aug 2022

Response to reviewer 2:

Reviewer # 2, Concern # 1: We confirm that all the images used in this work copyrights are free.

Author response: We have 

Reviewer # 2, Concern # 2: Also, the list of references should be carefully checked to ensure consistency with between all references and their compliances with the journal policy on referencing.

Author response: We have revised the format of the references to the style required by the journal.

---

## [Decision Letter · Decision Letter 3]

26 Aug 2022

Tax Preference, Financing Constraints and Enterprise Investment Efficiency——Experience of China’s Enterprises Investment

PONE-D-22-10870R3

Dear Dr. Feng,

We’re pleased to inform you that your manuscript has been judged scientifically suitable for publication and will be formally accepted for publication once it meets all outstanding technical requirements.

Kind regards,

Ali Safaa Sadiq

Academic Editor

PLOS ONE

Additional Editor Comments (optional):

I am happy to accept the manuscript as the authors could successfully addressed the given comments by the reviewers.

Reviewers' comments:

Reviewer's Responses to Questions

**Comments to the Author**

1. If the authors have adequately addressed your comments raised in a previous round of review and you feel that this manuscript is now acceptable for publication, you may indicate that here to bypass the “Comments to the Author” section, enter your conflict of interest statement in the “Confidential to Editor” section, and submit your "Accept" recommendation.

Reviewer #1: All comments have been addressed

Reviewer #2: (No Response)

2. Is the manuscript technically sound, and do the data support the conclusions?

Reviewer #1: Yes

Reviewer #2: (No Response)

3. Has the statistical analysis been performed appropriately and rigorously? 

Reviewer #1: Yes

Reviewer #2: (No Response)

4. Have the authors made all data underlying the findings in their manuscript fully available?

Reviewer #1: Yes

Reviewer #2: (No Response)

5. Is the manuscript presented in an intelligible fashion and written in standard English?

Reviewer #1: Yes

Reviewer #2: (No Response)

6. Review Comments to the Author

Reviewer #1: The authors have addressed all the comments and now paper is publishable in the journal. The authors have addressed all the comments and now paper is publishable in the journal.

Reviewer #2: Great work, all my comments have been addressed. Great work, all my comments have been addressed. Great work, all my comments have been addressed.

7. PLOS authors have the option to publish the peer review history of their article (what does this mean?). If published, this will include your full peer review and any attached files.

Reviewer #1: **Yes: **Ehsan Elahi

Reviewer #2: No

---

## [Editor Report · Acceptance letter]

9 Sep 2022

PONE-D-22-10870R3 

Tax Preference, Financing Constraints and Enterprise Investment Efficiency
——Experience of China’s Enterprises Investment 

Dear Dr. Feng:

I'm pleased to inform you that your manuscript has been deemed suitable for publication in PLOS ONE. Congratulations! Your manuscript is now with our production department. 

Kind regards, 

on behalf of

Dr. Ali Safaa Sadiq 

Academic Editor

PLOS ONE